# META-LEARNING AN INFERENCE ALGORITHM FOR PROBABILISTIC PROGRAMS

## ABSTRACT

We present a meta-algorithm for learning a posterior-inference algorithm for restricted probabilistic programs. Our meta-algorithm takes a training set of probabilistic programs that describe models with observations, and attempts to learn an efficient method for inferring the posterior of a similar program. A key feature of our approach is the use of what we call a white-box inference algorithm that extracts information directly from model descriptions themselves, given as programs. Concretely, our white-box inference algorithm is equipped with multiple neural networks, one for each type of atomic command, and computes an approximate posterior of a given probabilistic program by analysing individual atomic commands in the program using these networks. The parameters of the networks are learnt from a training set by our meta-algorithm. We empirically demonstrate that the learnt inference algorithm generalises well to programs that are new in terms of both parameters and model structures, and report cases where our approach achieves greater test-time efficiency than alternative approaches such as HMC. The overall results show the promise as well as remaining challenges of our approach.

## 1 INTRODUCTION

One key objective of probabilistic programming is to automate reasoning about probabilistic models from diverse domains (Ritchie et al., 2015; Perov & Wood, 2016; Baydin et al., 2019; Schaechtle et al., 2016; Cusumano-Towner et al., 2017; Saad & Mansinghka, 2016; Kulkarni et al., 2015; Young et al., 2019; Jäger et al., 2020). As a way to realize this goal, researchers have extensively worked on the development of posterior-inference or parameter-learning algorithms that are efficient and universal; the algorithms can be applied to all or nearly all models written in probabilistic programming languages (PPLs). This line of research has led to performant probabilistic programming systems (Goodman et al., 2008; Wood et al., 2014; Mansinghka et al., 2014; Minka et al., 2018; Narayanan et al., 2016; Salvatier et al., 2016; Carpenter et al., 2017; Tran et al., 2016; Ge et al., 2018; Bingham et al., 2018). Yet, it also revealed the difficulty of achieving efficiency and universality simultaneously, and the need for equipping PPLs with mechanisms for customising inference or learning algorithms to a given domain. In fact, recent PPLs include constructs for specifying conditional independence in a model (Bingham et al., 2018) or defining proposal or variational distributions (Ritchie et al., 2015; Siddharth et al., 2017; Bingham et al., 2018; Tran et al., 2018; Cusumano-Towner et al., 2019), all enabling users to help inference or learning algorithms.

In this paper, we explore a different approach. We present a meta-algorithm for learning a posterior-inference algorithm itself from a given set of restricted probabilistic programs, which specifies a class of probabilistic models, such as hierarchical or clustering models. The meta-algorithm aims at constructing a customised inference algorithm for the given set of models, while ensuring universality to the extent that the constructed algorithm can generalise: it works well for models not in the training set, as long as the models are similar to the ones in the set.

The distinguished feature of our approach is the use of what we call a white-box inference algorithm, which extracts information directly from model descriptions themselves, given as programs in a PPL. Concretely, our white-box inference algorithm is equipped with multiple neural networks, one for each type of atomic command in a PPL, and computes an approximate posterior for a given program by analysing (or executing in a sense) individual atomic commands in it using these networks. For instance, given the probabilistic program in Fig. 1, which describes a simple model on the Milky Way

$mass \sim \mathcal{N}(5, 10)$; // log of the mass of Milky Way
$g_1 \sim \mathcal{N}(mass \times 2, 5)$; $\mathtt{obs}(\mathcal{N}(g_1, 1), 10)$; // observed velocity $vel_1{=}10$ of the first satellite galaxy
$g_2 \sim \mathcal{N}(mass + 5, 2)$; $\mathtt{obs}(\mathcal{N}(g_2, 1), 3)$ // observed velocity $vel_2{=}3$ of the second satellite galaxy

Figure 1: Probabilistic program for a model for Milky Way and its two satellite galaxies. The $\mathtt{obs}$ statements refer to the observations of (unnamed) random variables $vel_1$ and $vel_2$.

galaxy, the white-box inference algorithm analyses the program as if an RNN handles a sequence or an interpreter executes a program. Roughly, the algorithm regards the program as a sequence of the five atomic commands (separated by the ";" symbol), initialises its internal state $h \in \mathbb{R}^m$ with $h_0$, and transforms the state over the sequence. The internal state $h$ is the encoding of an approximate posterior at the current program point, which corresponds to an approximate filtering distribution of a state-space model. How to update this state over each atomic command is directed by neural networks. Our meta-algorithm trains the parameters of these networks by trying to make the inference algorithm compute accurate posterior approximations over a training set of probabilistic programs. One can also view our white-box inference algorithm as a message-passing algorithm in a broad sense where transforming the internal state $h$ corresponds to passing a message, and understand our meta-algorithm as a method for learning how to pass a message for each type of atomic commands.

This way of exploiting model descriptions for posterior inference has two benefits. First, it ensures that even after customisation through the neural-network training, the inference algorithm does not lose its universality and can be applied to any probabilistic programs. Thus, at least in principle, the algorithm has a possibility to generalise beyond the training set; its accuracy degrades gracefully as the input probabilistic program diverges from those in the training set. Second, our way of using model descriptions guarantees the efficiency of the inference algorithm (although it does not guarantee the accuracy). The algorithm scans the input program only once, and uses neural networks whose input dimensions are linear in the size of the program. As a result, its time complexity is quadratic over the size of the input program. Of course, the guaranteed speed also indicates that the customisation of the algorithm for a given training set, whose main goal is to achieve good accuracy for probabilistic programs in the set, is a non-trivial process.

Our contributions are as follows: (i) we present a white-box posterior-inference algorithm, which works directly on model description and can be customised to a given model class; (ii) we describe a meta-algorithm for learning the parameters of the inference algorithm; (iii) we empirically analyse our approach with different model classes, and show the promise as well as the remaining challenges.

**Related work** The difficulty of developing an effective posterior-inference algorithm is well-known, and has motivated active research on learning or adapting key components of an inference algorithm. Techniques for adjusting an MCMC proposal (Andrieu & Thoms, 2008) or an HMC integrator (Hoffman & Gelman, 2014) to a given inference task were implemented in popular tools. Recently, methods for meta-learning these techniques themselves from a collection of inference tasks have been developed (Wang et al., 2018; Gong et al., 2019). The meta-learning approach also features in the work on stochastic variational inference where a variational distribution receives information about each inference task in the form of its dataset of observations and is trained with a collection of datasets (Wu et al., 2020; Gordon et al., 2019; Iakovleva et al., 2020). For a message-passing-style variational-inference algorithm, such as expectation propagation (Minka, 2001; Wainwright & Jordan, 2008), Jitkrittum et al. (2015) studied the problem of learning a mechanism to pass a message for a given *single* inference task. A natural follow-up question is how to meta-learn such a mechanism from a dataset of *multiple* inference tasks that can generalise to *unseen* models. Our approach provides a partial answer to the question; our white-box inference algorithm can be viewed as a message-passing-style variational inference algorithm that can meta-learn the representation of messages and a mechanism for passing them for given probabilistic programs.

Amortised inference and inference compilation (Gershman & Goodman, 2014; Le et al., 2017; Paige & Wood, 2016; Stuhlmüller et al., 2013; Kingma & Welling, 2013; Mnih & Gregor, 2014; Rezende et al., 2014; Ritchie et al., 2016; Marino et al., 2018) are closely related to our approach in that they also attempt to learn a form of a posterior-inference algorithm. However, the learnt algorithm by them and that by ours have different scopes. The former is designed to work for unseen inputs or observations of a *single* model, while the latter for *multiple* models with different structures. The relationship between these two algorithms is similar to the one between a compiled program (to be applied to multiple inputs) and a compiler (to be used for multiple programs).

$u := 0;\ v := 5;\ w := 1;\ z_1 \sim \mathcal{N}(u, v);\ z_2 \sim \mathcal{N}(u, v);$

$z_3 \sim \mathcal{N}(u, w);\ \mu_3 := \texttt{if}\ (z_3 > u)\ z_1\ \texttt{else}\ z_2;\ \texttt{obs}(\mathcal{N}(\mu_3, w), -1.9);\ \texttt{//}\ x_1 \sim \mathcal{N}(\mu_3, w), x_1 = -1.9$

$z_4 \sim \mathcal{N}(u, w);\ \mu_4 := \texttt{if}\ (z_4 > u)\ z_1\ \texttt{else}\ z_2;\ \texttt{obs}(\mathcal{N}(\mu_4, w), -2.2);\ \texttt{//}\ x_2 \sim \mathcal{N}(\mu_4, w), x_2 = -2.2$

$z_5 \sim \mathcal{N}(u, w);\ \mu_5 := \texttt{if}\ (z_5 > u)\ z_1\ \texttt{else}\ z_2;\ \texttt{obs}(\mathcal{N}(\mu_5, w), 2.4);\ \texttt{//}\ x_3 \sim \mathcal{N}(\mu_5, w), x_3 = 2.4$

$z_6 \sim \mathcal{N}(u, w);\ \mu_6 := \texttt{if}\ (z_6 > u)\ z_1\ \texttt{else}\ z_2;\ \texttt{obs}(\mathcal{N}(\mu_6, w), 2.2)\ \texttt{//}\ x_4 \sim \mathcal{N}(\mu_6, w), x_4 = 2.2$

Figure 2: Probabilistic program for a simple clustering model on four data points.

The idea of running programs with learnt neural networks also appears in the work on training neural networks to execute programs (Zaremba & Sutskever, 2014; Bieber et al., 2020; Reed & de Freitas, 2016). As far as we know, however, we are the first to frame the problem of learning a posterior-inference algorithm as the one of learning to execute.

## 2 SETUP

Our results assume a simple probabilistic programming language without loop and with a limited form of conditional statement. The syntax of the language is given by the following grammar, where $r$ represents a real number, $z$ and $v_i$ variables storing a real, and $p$ the name of a procedure taking two real-valued parameters and returning a real number:

$$\textit{Programs } C ::= A \mid C_1; C_2$$
$$\textit{Atomic Commands } A ::= z \sim \mathcal{N}(v_1, v_2) \mid \texttt{obs}(\mathcal{N}(v_0, v_1), r) \mid v_0 := \texttt{if}\ (v_1 > v_2)\ v_3\ \texttt{else}\ v_4$$
$$\mid\ v_0 := r \mid v_0 := v_1 \mid v_0 := p(v_1, v_2)$$

Programs in the language are constructed by sequentially composing atomic commands. The language supports six types of *atomic commands*. The first type is $z \sim \mathcal{N}(v_1, v_2)$, which draws a sample from the normal distribution with mean $v_1$ and variance $v_2$, and assigns the sampled value to $z$. The second command, $\texttt{obs}(\mathcal{N}(v_0, v_1), r)$, states that a random variable is drawn from $\mathcal{N}(v_0, v_1)$ and its value is observed to be $r$. The next is a restricted form of a conditional statement that selects one of $v_3$ and $v_4$ depending on the result of the comparison $v_1 > v_2$. The following two commands are different kinds of assignments, one for assigning a constant and the other for copying a value from one variable to another. The last atomic command $v_0 := p(v_1, v_2)$ is a call to one of the known deterministic procedures, which may be standard binary operations such as addition and multiplication, or complex non-trivial functions that are used to build advanced, non-conventional models. When $p$ is a standard binary operation, we use the usual infix notation and write, for example, $v_1 + v_2$, instead of $+(v_1, v_2)$.

We permit only the programs where a variable does not appear more than once on the left-hand side of the := and $\sim$ symbols. This means that no variable is updated twice or more, and it corresponds to the so-called static single assignment assumption in the work on compilers. This restriction lets us regard variables updated by $\sim$ as latent random variables. We denote those variables by $z_1, \ldots, z_n$.

We use this simple language for two reasons. First, the restriction imposed on our language enables the simple definition of our white-box inference algorithm. The language supports only a limited form of conditional statements and restricts the syntactic forms of atomic commands; the arguments to a normal distribution or to a procedure $p$ should be variables, not general expression forms such as addition of two variables. As we will show soon, this restriction makes it easy to exploit information about the type of each atomic command in our inference algorithm; we use different neural networks for different types of atomic commands in the algorithm. Second, the language is intended to serve as an intermediate language of a compiler for a high-level PPL, not the one to be used directly by the end user. The compilation scheme in, for instance, §3 of (van de Meent et al., 2018) from high-level probabilistic programs with general conditional statements and for loops to graphical models can be adopted to compile such programs into our language. See Appendix A for further discussion.

Fig. 2 shows a simple model for clustering four data points $\{-1.9, -2.2, 2.4, 2.2\}$ into two clusters, where the cluster assignment of each data point is decided by thresholding a sample from the standard normal distribution. The variables $z_1$ and $z_2$ store the centers of the two clusters, and $z_3, \ldots, z_6$ hold the random draws that decide cluster assignments for the data points. See Appendix B for the Milky Way example in Fig. 1 compiled to a program in our language.

Probabilistic programs in the language denote unnormalised probability densities over $\mathbb{R}^n$ for some $n$. Specifically, for a program $C$, if $z_1, \ldots, z_n$ are all the variables assigned by the sampling statements

$z_i \sim \mathcal{N}(\ldots)$ in $C$ in that order and $C$ contains $m$ observe statements with observations $r_1, \ldots, r_m$, then $C$ denotes an unnormalised density $p_C$ over the real-valued random variables $z_1, \ldots, z_n$: $p_C(z_{1:n}) = p_C(x_{1:m} = r_{1:m}|z_{1:n}) \times \prod_{i=1}^{n} p_C(z_i|z_{1:i-1})$, where $x_1, \ldots, x_m$ are variables not appearing in $C$ and are used to denote observed variables. This density is defined inductively over the structure of $C$. See Appendix C for details. The goal of our white-box inference algorithm is to compute efficiently accurate approximate posterior and marginal likelihood estimate for a given $C$ (that is, for the normalised version of $p_C$ and the normalising constant of $p_C$), when $p_C$ has a finite non-zero marginal likelihood and, as a result, a well-defined posterior density. We next describe how the algorithm attempts to achieve this goal.

## 3 WHITE-BOX INFERENCE ALGORITHM

Given a program $C = (A_1; \ldots; A_k)$, our white-box inference algorithm views $C$ as a sequence of its constituent atomic commands $(A_1, A_2, \ldots, A_k)$, and computes an approximate posterior and a marginal likelihood estimate for $C$ by sequentially processing the $A_i$'s. Concretely, the algorithm starts by initialising its internal state to $h_0 = \vec{0} \in \mathbb{R}^s$ and the current marginal-likelihood estimate to $Z_0 = 1$. Then, it updates these two components based on the first atomic command $A_1$ of $C$. It picks a neural network appropriate for the type of $A_1$, applies it to $h_0$ and gets a new state $h_1 \in \mathbb{R}^s$. Also, it updates the marginal likelihood estimate to $Z_1$ by analysing the semantics of $A_1$. This process is repeated for the remaining atomic commands $A_2, A_3, \ldots, A_k$ of $C$, and eventually produces the last state $h_k$ and estimate $Z_k$. Finally, the state $h_k$ gets decoded to a probability density on the latent variables of $C$ by a neural network, which together with $Z_k$ becomes the result of the algorithm.

Formally, our inference algorithm is built on top of three kinds of neural networks: the ones for transforming the internal state $h \in \mathbb{R}^s$ of the algorithm; a neural network for decoding the internal states $h$ to probability densities; and the last neural network for approximately solving integration questions that arise from the marginal likelihood computation in observe statements. We present these neural networks for the programs that sample $n$-many latent variables $z_1, \ldots, z_n$, and use at most $m$-many variables (so $m \geq n$). Let $\mathbb{V}$ be $[0, 1]^m$, the space of the one-hot encodings of those $m$ variables, and $\mathbb{P}$ the set of procedure names. Our algorithm uses the following neural networks:

$$nn_{\text{sa},\phi_1} : \mathbb{V}^3 \times \mathbb{R}^s \to \mathbb{R}^s, \qquad nn_{\text{ob},\phi_2} : \mathbb{V}^2 \times \mathbb{R} \times \mathbb{R}^s \to \mathbb{R}^s, \quad nn_{\text{if},\phi_3} : \mathbb{V}^5 \times \mathbb{R}^s \to \mathbb{R}^s,$$
$$nn_{:=,\phi_4}^{\text{c}} : \mathbb{V} \times \mathbb{R} \times \mathbb{R}^s \to \mathbb{R}^s, \qquad nn_{:=,\phi_5}^{\text{v}} : \mathbb{V}^2 \times \mathbb{R}^s \to \mathbb{R}^s, \qquad nn_{p,\phi_p} : \mathbb{V}^3 \times \mathbb{R}^s \to \mathbb{R}^s \text{ for } p \in \mathbb{P},$$
$$nn_{\text{de},\phi_6} : \mathbb{R}^s \to (\mathbb{R} \times \mathbb{R})^n, \qquad nn_{\text{intg},\phi_7} : \mathbb{V}^2 \times \mathbb{R} \times \mathbb{R}^s \to \mathbb{R},$$

where $\phi_{1:7}$ and $\phi_p$ for $p \in \mathbb{P}$ are network parameters. The top six networks are for the six types of atomic commands in our language. For instance, when an atomic command to analyse next is a sample statement $z \sim \mathcal{N}(v_1, v_2)$, the algorithm runs the first network $nn_{\text{sa}}$ on the current internal state $h$, and obtains a new state $h' = nn_{\text{sa},\phi_1}(\overline{z}, \overline{v_{1:2}}, h)$, where $\overline{z}$ and $\overline{v_{1:2}}$ mean the one-hot encoded variables $z$, $v_1$ and $v_2$. The next $nn_{\text{de},\phi_6}$ is a decoder of the states $h$ to probability densities over the latent variables $z_1, \ldots, z_n$, which are the product of $n$ independent normal distributions. The network maps $h$ to the means and variances of these distributions. The last $nn_{\text{intg},\phi_7}$ is used when our algorithm updates the marginal likelihood estimate based on an observe statement $\text{obs}(\mathcal{N}(v_0, v_1), r)$. When we write the meaning of this observe statement as the likelihood $\mathcal{N}(r; v_0, v_1)$, and the filtering distribution for $v_0$ and $v_1$ under (the decoded density of) the current state $h$ as $p_h(v_0, v_1)$,[1] the last neural network computes the following approximation: $nn_{\text{intg},\phi_7}(\overline{v_{0:1}}, r, h) \approx \int \mathcal{N}(r; v_0, v_1) p_h(v_0, v_1) dv_0 dv_1$. See Appendix D for the full derivation of the marginal likelihood.

Given a program $C = (A_1; \ldots; A_k)$ that draws $n$ samples (and so uses latent variables $z_1, \ldots, z_n$), the algorithm approximates the posterior and marginal likelihood of $C$ as follows:

$$\text{INFER}(C) = \textbf{let } (h_0, Z_0) = (\vec{0}, 1) \textbf{ and } (h_k, Z_k) = (\text{INFER}(A_k) \circ \ldots \circ \text{INFER}(A_1))(h_0, Z_0) \textbf{ in}$$
$$\textbf{let } ((\mu_1, \sigma_1^2), \ldots, (\mu_n, \sigma_n^2)) = nn_{\text{de},\phi_6}(h_k) \textbf{ in return } \Big( \prod_{i=1}^{n} \mathcal{N}(z_i \mid \mu_i, \sigma_i^2), \; Z_k \Big),$$

where $\text{INFER}(A_i) : \mathbb{R}^s \times \mathbb{R} \to \mathbb{R}^s \times \mathbb{R}$ picks an appropriate neural network based on the type of $A_i$, and uses it to transform $h$ and $Z$:

$$\text{INFER}(\text{obs}(\mathcal{N}(v_0, v_1), r))(h, Z) = (nn_{\text{ob}}(\overline{v_{0:1}}, r, h), \; Z \times nn_{\text{intg}}(\overline{v_{0:1}}, r, h)),$$

---

[1] The $p_h(v_0, v_1)$ is a filtering distribution, not prior.

$\text{INFER}(v_0 := \texttt{if } (v_1 > v_2) \; v_3 \; \texttt{else} \; v_4)(h, Z) = (nn_{\text{if}}(\overline{v_{0:4}}, h), Z),$

$\text{INFER}(v_0 := r)(h, Z) = (nn_{:=}^{\text{c}}(\overline{v_0}, r, h), Z), \; \text{INFER}(z \sim \mathcal{N}(v_1, v_2))(h, Z) = (nn_{\text{sa}}(\overline{z}, \overline{v_{1:2}}, h), Z),$

$\text{INFER}(v_0 := v_1)(h, Z) = (nn_{:=}^{\text{v}}(\overline{v_{0:1}}, h), Z), \; \text{INFER}(v_0 := p(v_1, v_2))(h, Z) = (nn_p(\overline{v_{0:2}}, h), Z).$

We remind the reader that $\overline{v_{0:k}}$ refers to the sequence of the one-hot encodings of variables $v_0, \dots, v_k$. For the update of the state $h$, the subroutine $\text{INFER}(A)$ relies on neural networks. But for the computation of the marginal likelihood estimate, it exploits prior knowledge that non-observe commands do not change the marginal likelihood (except only indirectly by changing the filtering distribution), and keeps the input $Z$ for those atomic commands.

## 4 META-LEARNING PARAMETERS

The parameters of our white-box inference algorithm are learnt from a collection of probabilistic programs in our language. Assume that we are given a training set of programs $\mathcal{D} = \{C_1, \dots, C_N\}$ such that each $C_i$ samples $n$ latent variables $z_1, \dots, z_n$ and uses at most $m$ variables. Let $\phi = (\phi_{1:7}, (\phi_p)_{p \in \mathbb{P}})$ be the parameters of all the neural networks used in the algorithm. We learn these parameters by solving the following optimisation problem:[2]

$$\arg\min_\phi \sum_{C \in \mathcal{D}} \text{KL}[\pi_C(z_{1:n}) || q_C(z_{1:n})] + \frac{\lambda}{2}(N_C - Z_C)^2$$

where $\lambda > 0$ is a hyper-parameter, $N_C$ is the marginal likelihood (or the normalising constant) $\int p_C(z_{1:n}) dz_{1:n}$ for $p_C$, the next $\pi_C(z_{1:n})$ is the normalised posterior $p_C(z_{1:n})/N_C$ for $C$, and the last $q_C$ and $Z_C$ are the approximate posterior and marginal likelihood estimate computed by the inference algorithm (that is, $(q_C(z_{1:n}), Z_C) = \text{INFER}(C)$). Note that $q_C$ and $Z_C$ both depend on $\phi$, since INFER uses the $\phi$-parameterised neural networks.

We optimise the objective by stochastic gradient descent. The key component of the optimisation is a gradient estimator derived as follows: $(\nabla_\phi \sum_{C \in \mathcal{D}} \text{KL}[\pi_C || q_C] + \frac{\lambda}{2}(N_C - Z_C)^2) = (\sum_{C \in \mathcal{D}} \mathbb{E}_{z_{1:n} \sim \pi_C}[-\nabla_\phi \log q_C(z_{1:n})] - \lambda(N_C - Z_C)\nabla_\phi Z_C) \approx \sum_{C \in \mathcal{D}} -\widehat{L_{C,\phi}} - \lambda(\widehat{N_C} - Z_C)\nabla_\phi Z_C$. Here $\widehat{L_{C,\phi}}$ and $\widehat{N_C}$ are sample estimates of $\mathbb{E}_{z_{1:n} \sim \pi_C}[\nabla_\phi \log q_C(z_{1:n})]$ and the marginal likelihood, respectively. Both estimates can be computed using standard Monte-Carlo algorithms. For instance, we can run the self-normalising importance sampler with prior as proposal, and generate weighted samples $\{(w^{(j)}, z_{1:n}^{(j)})\}_{1 \le j \le M}$ for the unnormalised posterior $p_C$. Then, we can use these samples to compute the required estimates: $\widehat{N_C} = \frac{1}{M} \sum_{j=1}^M w^{(j)}$ and $\widehat{L_{C,\phi}} = \frac{1}{M} \sum_{j=1}^M (w^{(j)} \nabla_\phi \log q_C(z_{1:n}^{(j)}))/\widehat{N_C}$. Alternatively, we may run Hamiltonian Monte Carlo (HMC) (Duane et al., 1987) to generate posterior samples, and use those samples to draw weighted importance samples using, for instance, the layered adaptive importance sampler (Martino et al., 2017). Then, we compute $\widehat{L_{C,\phi}}$ using posterior samples, and $\widehat{N_C}$ using weighted importance samples. Note that neither $\pi_C$ in $\mathbb{E}_{z_{1:n} \sim \pi_C}[-\nabla_\phi \log q_C(z_{1:n})]$ nor $N_C$ depends on the parameters $\phi$. Thus, for each $C \in \mathcal{D}$, $N_C$ needs to be estimated only once throughout the entire optimisation process, and the posterior samples from $\pi_C$ need to be generated only once as well. We use this fact to speed up the computation of each gradient-update step.

## 5 EMPIRICAL EVALUATION

An effective meta-algorithm should generalise well: the learnt inference algorithm should accurately predict the posteriors of programs unseen during training which have different parameters (§5.1) and model structures (§5.2), as long as the programs are similar to those in the training set. We empirically show that our meta-algorithm learns such an inference algorithm, and that in some cases using the learnt inference algorithm achieves higher test-time efficiency than alternative approaches such as HMC (Duane et al., 1987) (§5.3). We implemented our inference algorithm and meta-algorithm using ocaml-torch (Mazare, 2018), an OCaml binding for PyTorch. For HMC, we used the Python interface for Stan (Carpenter et al., 2017). We used a Ubuntu server with Intel(R) Xeon(R) Gold 6234 CPU @ 3.30GHz with 16 cores, 32 threads, and 263G memory. See Appendix E for the full list of our model classes and their details, and Appendix F for the detailed experimental setup.

---

[2]Strictly speaking, we assume that the marginal likelihood of any $C \in \mathcal{D}$ is non-zero and finite.

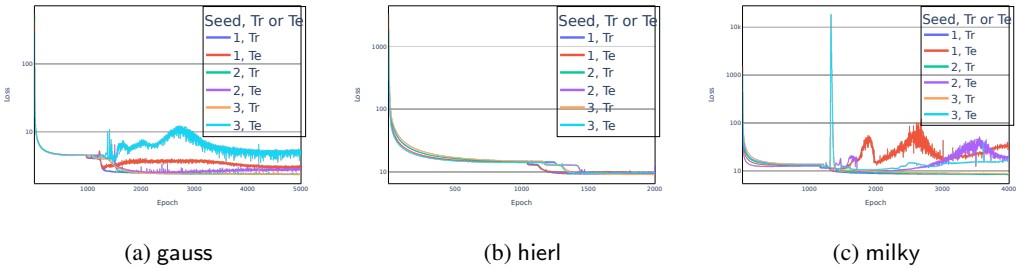

|  (a) gauss | (b) hierl | (c) milky |

Figure 3: Average training and test losses under three random seeds. The $y$-axes are log-scaled. The increases in later epochs of Fig. 3c were due to only one or a few test programs out of $50$.

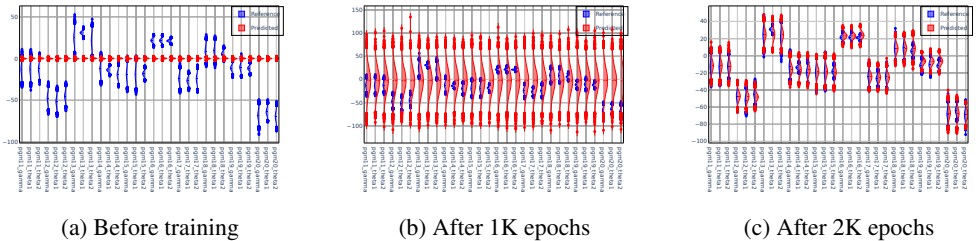

|  (a) Before training | (b) After 1K epochs | (c) After 2K epochs |

Figure 4: Comparisons of predicted and reference marginal posteriors recorded at different training steps: at the initial step, after 1K epochs, and after 2K epochs.

### 5.1 GENERALISATION TO NEW MODEL PARAMETERS AND OBSERVATIONS

We evaluated our approach with six model classes: (1) Gaussian models (gauss) with a latent variable and an observation where the mean of the Gaussian likelihood is an affine transformation of the latent; (2) hierarchical models with three hierarchically structured latent variables (hierl); (3) hierarchical or multi-level models with both latent variables and data structured hierarchically (hierd) where data are modelled as a regression of latent variables at different levels; (4) clustering models (cluster) where five observations are clustered into two groups; (5) Milky Way models (milky), and their multiple-observations extension (milkyo) where five observations are made for each satellite galaxy; and (6) models (rb) using the Rosenbrock function,[3] which is expressed as an external procedure, to show that our approach can in principle handle models with non-trivial computation blocks.

The purpose of our evaluation is to show the feasibility of our approach, not to develop the state-of-the-art inference algorithm automatically, and also to identify the challenges of the approach. These models are chosen for this purpose. For instance, an inference algorithm should be able to reason about affine transformations and Gaussian distributions (for gauss), and dependency relationships among variables (for hierl and hierd) to compute a posterior accurately. Successful outcomes in the classes indicate that our approach learns an inference algorithm with such capacity in some cases.

**Setup** For each model class, we used $400$ programs to meta-learn an inference algorithm, and then applied the learnt algorithm to $50$ unseen test programs. We measured the average test loss over the $50$ test programs, and checked if the loss also decreases when the training loss decreases. We also compared the marginal posteriors predicted by our learnt inference algorithm with the reference marginal posteriors that were computed analytically, or approximately by HMC. When we relied on HMC, we computed the marginal sample means and standard deviations using one of the $10$ Markov chains generated by independent HMC runs. Each chain consisted of 500K samples after 50K warmups. We ensured the convergence of the chains using diagnostics such as $\hat{R}$ (Gelman et al., 1992). All training and test programs were automatically generated by a random program generator. This generator takes a program class and hyperparameters (e.g., boundaries of the quantities that are used to specify the models), and returns programs from the class randomly (see Appendix E).

For each training program, our meta-algorithm used $2^{15}$ samples from the analytic (for gauss) or approximate (for the rest, by HMC) posterior distribution for the program.[4] Similarly, our meta-

---

[3]The function is often used to evaluate learning and inference algorithms (Goodman & Weare, 2010; Wang & Li, 2018; Pagani et al., 2019)

[4]Except for rb; see the discussion on Rosenbrock models in Appendix H.

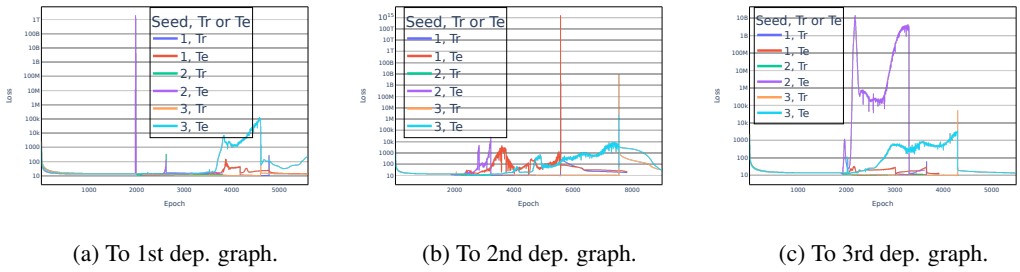

(a) To 1st dep. graph.  (b) To 2nd dep. graph.  (c) To 3rd dep. graph.

Figure 5: Average losses for generalisation to dependency graphs in ext1. The y-axes are log-scaled.

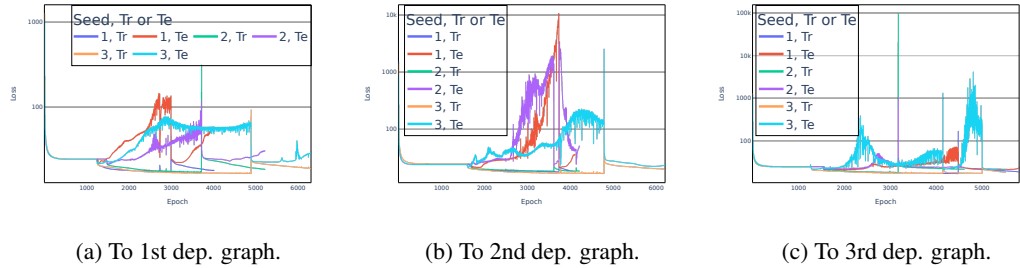

(a) To 1st dep. graph.  (b) To 2nd dep. graph.  (c) To 3rd dep. graph.

Figure 6: Average losses for generalisation to dependency graphs in ext2. The y-axes are log-scaled.

algorithm computed the marginal likelihood analytically (for gauss) or approximately (for the rest) using layered adaptive importance sampling (Martino et al., 2017) where the proposals were defined by an HMC chain. We performed mini-batch training; a single gradient update was done with a training program and a mini batch of size $2^{12}$ (out of $2^{15}$ samples for the program). We used Adam (Kingma & Ba, 2015) with its hyperparameters $\{\beta_1 = 0.9, \beta_2 = 0.999, \text{weight\_decay} = 0\}$, and the initial learning rate was set to $0.001$. When the average training loss converged enough, the training stopped. We repeated the same experiments three times using different random seeds.

**Results** Fig. 3 shows the training and test losses for gauss, hierl, and milky under three random seeds. The losses for the other model classes are in Appendix G. The training loss was averaged over the training set and $8$ batch updates, and the test loss over the test set. The plots for training and test losses are drawn as solid and dotted lines, respectively, and the results with different random seeds are coloured differently. The training losses in all three experiments decreased rapidly, and more importantly, these decreases were accompanied by the downturns of the test losses, which shows that the learnt parameters generalised to the test programs well. The later part of Fig. 3c shows cases where the test loss increases. This was because the loss of only a few programs in the test set (of $50$ programs) became large. Even in this situation, the losses of the rest remained small.

Fig. 4 compares, for $10$ test programs in hierl, the reference marginal posteriors (blue) and their predicted counterparts (red) by the learnt inference algorithm instantiated at three different training epochs. The predicted marginals were initially around zero (Fig. 4a), evolved to cover the reference marginals (Fig. 4b), and finally captured them precisely in terms of both mean and standard deviation for most of the variables (Fig. 4c). The results show that our meta-algorithm improves the parameters of our inference algorithm, and eventually finds optimal ones that generalise well. We observed similar patterns for the other model classes and random seeds, except for cluster and rb; programs from these classes often have multimodal posteriors, and we provide an analysis for them in Appendix H.

## 5.2 GENERALISATION TO NEW MODEL STRUCTURES

We let two kinds of model structure vary across programs: the dependency (or data-flow) graph for the variables of a program and the position of a nonlinear function in the program. Specifically, we considered two model classes: (1) models (ext1) with three Gaussian variables and one deterministic variable storing the value of the function $\text{nl}(x) = 50/\pi \times \arctan(x/10)$, where the models have 12 different types — four different dependency graphs of the variables, and three different positions of the deterministic nl variable for each of these graphs; and (2) models (ext2) with six Gaussian variables and one nl variable, which are grouped into five types based on their dependency graphs. The evaluation was done for ext1 and ext2 independently as in §5.1, but here each of ext1 and ext2

Table 1: ESS per sec for the 60 test programs by HMC vs. IS-pred vs. IS-prior.

| | HMC | | | IS-pred | | | IS-prior | | |
|---|---|---|---|---|---|---|---|---|---|
| | GM | Q1 | Q3 | GM | Q1 | Q3 | GM | Q1 | Q3 |
| ESS | 204.8K | 4.1K | 4.6M | 4.2K | 2.2K | 13.8K | 2.8K | 1.1K | 9.5K |
| Time | 48.2s | 27.4s | 82.3s | 22.7ms | 21.4ms | 23.0ms | 23.1ms | 22.1ms | 24.0ms |
| ESS / sec | 4.3K | 124 | 127.7K | **196.5K** | **102.4K** | **646.5K** | 123.8K | 52.6K | 436.1K |

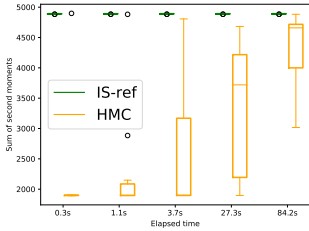
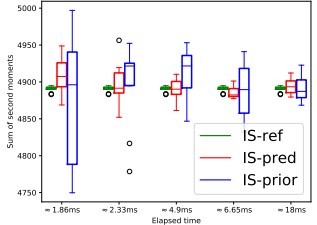
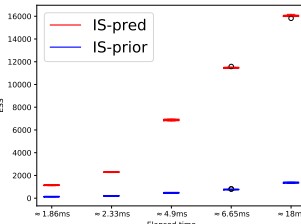

(a) Moments by HMC.  (b) Moments by IS-pred and IS-prior.  (c) ESS by IS-pred and IS-prior.

Figure 7

itself has programs of multiple (12 for ext1 and 5 for ext2) model types. See Fig. 10 and 11 in the appendix for visualisation of the different model types in ext1 and ext2, respectively.

**Setup** For ext1, we ran seven different experiments. Three of them evaluated generalisation to unseen positions of the nl variable, and the other four to unseen dependency graphs. Let $T_{i,j}$ be the type in ext1 that corresponds to the $i$-th position of nl and the $j$-th dependency graph, and $T_{-i,*}$ be all the types in ext1 that correspond to any nl positions except the $i$-th and any of four dependency graphs. For generalisation to the $i$-th position of nl ($i = 1, 2, 3$), we used programs from $T_{-i,*}$ for training and those from $T_{i,*}$ for testing. For generalisation to the $j$-th dependency graph ($j = 1, 2, 3, 4$), we used programs from $T_{*,-j}$ for training and those from $T_{*,j}$ for testing. For ext2, we ran five different experiments where each of them tested generalisation to an unseen dependency graph after training with the other four dependency graphs. All these experiments were repeated three times under different random seeds. So, the total numbers of experiment runs were $21 (= 7 \times 3)$ and $15 (= 5 \times 3)$ for ext1 and ext2, respectively.

In each experiment run for ext1, we used 720 programs for training, and 90 (when generalising to new graphs) or 100 (when generalising to new positions of the nl variable) unseen programs for testing. In each run for ext2, we used 600 programs for training and tested the learnt inference algorithm on 50 unseen programs. We ran HMC to estimate posteriors and marginal likelihoods, and used 200K samples after 10K warmups to compute reference posteriors. We stopped training after giving enough time for convergence within a limit of computational resources. The rest was the same as in §5.1.

**Results** Fig. 5 shows the average training and test losses for generalisation to the first three dependency graphs in ext1. Fig. 6 shows the losses for generalisation to the first three dependency graphs in ext2. The losses for generalisation to the last dependency graph and to all positions of the nl variable in ext1, and those for generalisation to the 4th and 5th dependency graphs in ext2 are in Appendices I and J. In 17 runs (out of 21) for ext1, the decrease in the training losses eventually stabilised or reduced the test losses, even when the test losses were high and fluctuated in earlier training epochs. In 8 runs (out of 15) for ext2, the test losses were stabilised as the training losses decreased. In 4 runs out of the other 7, the test losses increased only slightly. In terms of predicted posteriors, we observed highly accurate predictions in 8 runs for ext1. For ext2, the predicted posteriors were accurate in 7 runs. For quantified accuracy, we refer the reader to Appendix K. Overall, the learnt algorithms generalised to unseen types of models well or fairly well in many cases.

### 5.3 TEST-TIME EFFICIENCY IN COMPARISON WITH ALTERNATIVES

We demonstrate the test-time efficiency of our approach using three-variable models (mulmod) where two latent variables follow normal distributions and the other stores the value of the function $\mathrm{mm}(x) = 100 \times x^3/(10 + x^4)$. The models are grouped into three types defined by their dependency graphs and the positions of mm in the programs (see Fig.12 in the appendix). We ran our meta-algorithm using 600 programs from all three types using importance samples (not HMC samples).

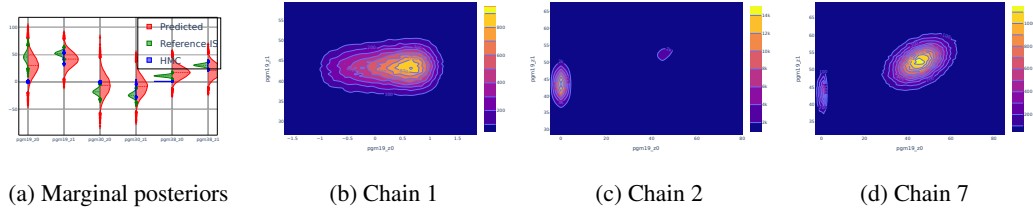

| (a) Marginal posteriors | (b) Chain 1 | (c) Chain 2 | (d) Chain 7 |

Figure 8: Marginal posteriors for the comparison, and contours of three HMC chains for pgm_19 where the x-axis is for z0 and the y-axis z1.

Then for 60 test programs from the last model type, we measured ESS and the sum of second moments along the wall-clock time using three approaches: importance sampling (IS-pred; ours) with the predicted posteriors as proposal using 70K samples, importance sampling (IS-prior) with prior as proposal using 100K samples, and HMC with 1M samples after 500 warmups. As the reference sampler, we used importance sampling (IS-ref) with prior as proposal using 5M samples. All the approaches were repeated 10 times.

Table 1 shows the average ESS per unit time over the 60 test programs, by the three approaches. For HMC, "ESS" is the ESS computed using 10 Markov chains averaged over the 60 programs, and "ESS / sec" is the ESS per unit time, averaged over the programs. For IS-{pred, prior}, "ESS" and "ESS / sec" are the average ESS and ESS per unit time, respectively, both over the 10 trials and the 60 programs. GM is the geometric mean, and Q1 and Q3 are the first and third quartiles, respectively. We used the geometric mean, since the ESSes had outliers. The results show that IS-pred achieved the highest ESS per unit time in terms of both mean (GM) and the quartiles (Q1 and Q3).

We provide further analysis for a test program (pgm19; see Appendix L). Fig. 7a and 7b show the moments estimated by HMC and IS-{pred, prior}, respectively, in comparison with the same (across the two figures) reference moments by IS-ref. The estimates by IS-pred (red) quickly converged to the reference (green) within 18ms, while those by HMC (orange) did not converge even after 84s. IS-pred (red) and IS-prior (blue) tended to produce better estimates as the elapsed time increased, but each time, IS-pred estimated the moments more precisely with a smaller variance than IS-prior. In the same runs of the three approaches as in the last columns of Fig. 7a and 7b, IS-pred produced over 16K effective samples in 18ms, while HMC generated only 80 effective samples even after 84s. Similarly, IS-prior generated fewer than 1.4K effective samples in the approximately same elapsed time as in IS-pred. In fact, Fig. 7c shows that as the time increases, the gap between the ESSes of IS-pred and IS-prior gets widen, because the former increases at a rate significantly higher than the latter. Note that IS-pred has to scan a program twice at test time, once for computing the proposal and another for IS with the predicted proposal. See Appendix M for discussion.

Our manual inspection revealed that the programs in mulmod often have multimodal posteriors. Fig. 8a shows the posteriors for {pgm19, pgm30, pgm38} in the test set, computed by our learnt inference algorithm (without IS), HMC (200K samples after 10K warmups) , and IS-ref. The variable z0 in the three programs had multimodal posteriors. For pgm19, the learnt inference algorithm took only 0.6ms to compute the posteriors, while HMC took 120s on average to generate a chain. The predictions (red) from the learnt inference algorithm for z0 describe the reference posteriors (green) better than those (blue) by HMC in terms of mean, variance, and mode covering.[5] The contour plots in Fig. 8b to 8d visualise three HMC chains for pgm19. Here, HMC failed to converge, and Fig. 8b explains the poor estimate (blue) in the first column of Fig. 8a.

**Conclusion** In this paper, we presented a white-box inference algorithm that computes an approximate posterior and a marginal likelihood estimate by analysing the given program sequentially using neural networks, and a meta-algorithm that learns the network parameters over a training set of probabilistic programs. In our experiments, the meta-algorithm learnt an inference algorithm that generalises well to similar but unseen programs, and the learnt inference algorithm sometimes had test-time advantages over alternatives. A moral of this work is that the description of a probabilistic model itself has useful information, and learning to extract and exploit the information may lead to an efficient inference. We hope that our work encourages further exploration of this research direction.

---

[5]Our inference algorithm in a multimodal-posterior case leads to a good approximation in the following sense: the approximating distribution $q$ covers the regions of the modes well, and also approximates the mean and variance of the target distribution accurately. Note that such a $q$ is useful when it is used as the proposal of an importance sampler.

**Reproducibility statement** Our paper provides detailed information that is needed for reproducing the results. In each of §5.1, §5.2, and §5.3 of the main text, we explain the key experimental design and setup clearly. More detailed experimental setup is in Appendix F, where we specify the hyperparameter and the design of the neural networks. In Appendix E, we provide full details of the model classes and how programs from the classes were automatically generated in our evaluation.

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

## A    FURTHER DISCUSSION ABOUT THE TRANSLATION OF AN EXPRESSIVE PPL INTO OUR INTERMEDIATE LANGUAGE

Programs with recursion or while loops cannot generally be translated into our intermediate language, since such programs may go into infinite loops while the programs in our language always terminate. Programs with for loops and general branches can in theory be translated into a less expressive language such as ours. For example, van de Meent et al. (2018) explain a language called FOPPL (Section 2), which has for loops and branches, and the translation of FOPPL into graphical models (Section 3). We think that these graphical models can be translated into programs in our language. Of course, this does not mean that the learnt inference algorithm would interact well with the compilation; the interaction between compilation and inference in the context of meta-learning is something to be explored in future work.

---

$one := 1; \ t := 2; \ f := 5; \ ten := 10;$

$z_1 \sim \mathcal{N}(f, ten); \ // \ \text{log of the mass of Milky Way}$

$mass_1 := z_1 \times t;$

$z_2 \sim \mathcal{N}(mass_1, f); \ // \ \text{for the first satellite galaxy}$

$\texttt{obs}(\mathcal{N}(z_2, one), ten); \ // \ x_1 = 10 \text{ for } x_1 \sim \mathcal{N}(z_2, one)$

$mass_2 := z_1 + f;$

$z_3 \sim \mathcal{N}(mass_2, t); \ // \ \text{for the second satellite galaxy}$

$\texttt{obs}(\mathcal{N}(z_3, one), 3) \ // \ x_2 = 3 \text{ for } x_2 \sim \mathcal{N}(z_3, one)$

---

Figure 9: Milky Way example compiled to the probabilistic programming language used in the paper.

## B    MILKY WAY EXAMPLE IN THE PROBABILISTIC PROGRAMMING LANGUAGE

Fig. 9 shows the compiled version of the Milky way example to the probabilistic programming language of the paper.

## C    FORMAL SEMANTICS OF THE PROBABILISTIC PROGRAMMING LANGUAGE

In §2, we stated that a program $C$ in our language denotes an unnormalised density $p_C$ that is factorised as follows:

$$p_C(z_{1:n}) = p_C(x_{1:m} = r_{1:m} | z_{1:n}) \times \prod_{i=1}^{n} p_C(z_i | z_{1:i-1}).$$

Here $z_1, \ldots, z_n$ are all the variables assigned by the sampling statements $z_i \sim \mathcal{N}(\ldots)$ in $C$ in that order, the program $C$ contains $m$ observe statements with observations $r_1, \ldots, r_m$, and these observed random variables are denoted by $x_1, \ldots, x_m$. The goal of this section is to provide the details of our statement. That is, we describe the formal semantics of our probabilistic programming language, and from it, we derive a map from programs $C$ to unnormalised densities $p_C$.

To define the formal semantics of programs in our language, we need a type system that tracks information about updated variables and observations, and also formalises the syntactic conditions that we imposed informally in §2. The type system lets us derive the following judgements for programs $C$ and atomic commands $A$:

$$(S, V, \alpha) \vdash_1 C : (T, W, \beta), \quad (S, V, \alpha) \vdash_2 A : (T, W, \beta),$$

where $S$ and $T$ are sequences of distinct variables, $V$ and $W$ are sets of variables that do not appear in $S$ and $T$, respectively, and $\alpha$ and $\beta$ are sequences of reals. The first judgement says that if before running the program $C$, the latent variables in $S$ are sampled in that order, the program variables in $V$ are updated by non-sample statements, and the real values in the sequence $\alpha$ are observed in that order, then running $C$ changes these three data to $T$, $W$, and $\beta$. The second judgement means

the same thing except that we consider the execution of $A$, instead of $C$. The triples $(S, V, \alpha)$ and $(T, W, \beta)$ serve as types in this type system.

The rules for deriving the judgements for $C$ and $A$ follow from the intended meaning just explained. We show these rules below, using the notation @ for the concatenation operator for two sequences and also $\mathrm{set}(S)$ for the set of elements in the sequence $S$:

$$\frac{(R, U, \alpha) \vdash_1 C_1 : (S, V, \beta) \quad (S, V, \beta) \vdash_1 C_2 : (T, W, \gamma)}{(R, U, \alpha) \vdash_1 (C_1; C_2) : (T, W, \gamma)} \qquad \frac{(S, V, \alpha) \vdash_2 A : (T, W, \beta)}{(S, V, \alpha) \vdash_1 A : (T, W, \beta)}$$

$$\frac{z \notin \mathrm{set}(S) \cup V \quad v_1, v_2 \in \mathrm{set}(S) \cup V}{(S, V, \alpha) \vdash_2 (z \sim \mathcal{N}(v_1, v_2)) : (S@[z], V, \alpha)} \qquad \frac{v_0, v_1 \in \mathrm{set}(S) \cup V}{(S, V, \alpha) \vdash_2 \mathtt{obs}(\mathcal{N}(v_0, v_1), r) : (S, V, \alpha@[r])}$$

$$\frac{v_0 \notin \mathrm{set}(S) \cup V \quad v_1, v_2, v_3, v_4 \in \mathrm{set}(S) \cup V}{(S, V, \alpha) \vdash_2 (v_0 := \mathtt{if}\ (v_1 > v_2)\ v_3\ \mathtt{else}\ v_4) : (S, V \cup \{v_0\}, \alpha)}$$

$$\frac{v_0 \notin \mathrm{set}(S) \cup V}{(S, V, \alpha) \vdash_2 (v_0 := r) : (S, V \cup \{v_0\}, \alpha)} \qquad \frac{v_0 \notin \mathrm{set}(S) \cup V \quad v_1 \in \mathrm{set}(S) \cup V}{(S, V, \alpha) \vdash_2 (v_0 := v_1) : (S, V \cup \{v_0\}, \alpha)}$$

$$\frac{v_0 \notin \mathrm{set}(S) \cup V \quad v_1, v_2 \in \mathrm{set}(S) \cup V}{(S, V, \alpha) \vdash_2 (v_0 := p(v_1, v_2)) : (S, V \cup \{v_0\}, \alpha)}$$

We now define our semantics, which specifies mappings from judgements for $C$ and $A$ to mathematical entities. First, we interpret each type $(S, V, \alpha)$ as a set, and it is denoted by $[\![(S, V, \alpha)]\!]$:

$$[\![(S, V, \alpha)]\!] = \{(p, f, l) \mid p \text{ is a (normalised) density on } \mathbb{R}^{|S|}, \ f = (f_v)_{v \in \mathrm{set}(S) \cup V},$$

$$\text{each } f_v \text{ is a measurable map from } \mathbb{R}^{|S|} \text{ to } \mathbb{R},$$

$$l \text{ is a measurable function from } \mathbb{R}^{|S|} \times \mathbb{R}^{|\alpha|} \text{ to } \mathbb{R}_+\},$$

where $|S|$ and $|\alpha|$ are the lengths of the sequences $S$ and $\alpha$, and $\mathbb{R}_+$ means the set of positive reals. Next, we define the semantics of the judgements $(S, V, \alpha) \vdash_1 C : (T, W, \beta)$ and $(S, V, \alpha) \vdash_2 A : (T, W, \beta)$ that can be derived by the rules from above. The formal semantics of these judgements, denoted by the $[\![-]\!]$ notation, are maps of the following type:

$$[\![(S, V, \alpha) \vdash_1 C : (T, W, \beta)]\!] : [\![(S, V, \alpha)]\!] \to [\![(T, W, \beta)]\!],$$
$$[\![(S, V, \alpha) \vdash_2 A : (T, W, \beta)]\!] : [\![(S, V, \alpha)]\!] \to [\![(T, W, \beta)]\!].$$

The semantics is given by induction on the size of the derivation of each judgement, under the assumption that for each procedure name $p \in \mathbb{P}$, we have its interpretation as a measurable map from $\mathbb{R}^2$ to $\mathbb{R}$:

$$[\![p]\!] : \mathbb{R}^2 \to \mathbb{R}.$$

We spell out the semantics below, first the one for programs and next that for atomic commands.

$$[\![(S, V, \alpha) \vdash_1 A : (T, W, \beta)]\!](p, f, l) = [\![(S, V, \alpha) \vdash_2 A : (T, W, \beta)]\!](p, f, l),$$
$$[\![(R, U, \alpha) \vdash_1 (C_1; C_2) : (T, W, \gamma)]\!](p, f, l) = ([\![(S, V, \beta) \vdash_2 C_2 : (T, W, \gamma)]\!]$$
$$\circ\, [\![(R, U, \alpha) \vdash_2 C_1 : (S, V, \beta)]\!])(p, f, l).$$

Let $\mathcal{N}(a; b, c)$ be the density of the normal distribution with mean $b$ and variance $c$ when $c > 0$ and 1 when $c \le 0$. For a family of functions $f = (f_v)_{v \in V}$, a variable $w \notin V$, and a function $f'_w$, we write $f \oplus f'_w$ for the extension of $f$ with a new $w$-indexed member $f'_w$.

$$[\![(S, V, \alpha) \vdash_2 z \sim \mathcal{N}(v_1, v_2) : (S@[z], V, \alpha)]\!](p, f, l) = (p', f', l')$$

$$\text{(where } p'(a_{1:|S|+1}) = p(a_{1:|S|}) \times \mathcal{N}(a_{|S|+1}; f_{v_1}(a_{1:|S|}), f_{v_2}(a_{1:|S|})),$$

$$f'_v(a_{1:|S|+1}) = f_v(a_{1:|S|}) \text{ for all } v \in V, \ f'_z(a_{1:|S|+1}) = a_{|S|+1},$$

$$l'(a_{1:|S|+1}, b_{1:|\alpha|}) = l(a_{1:|S|}, b_{1:|\alpha|})),$$

$$[\![(S, V, \alpha) \vdash_2 \mathtt{obs}(\mathcal{N}(v_0, v_1), r) : (S, V, \alpha@[r])]\!](p, f, l) = (p, f, l')$$

$$(\text{where } l'(a_{1:|S|}, b_{1:|\alpha|+1}) = l(a_{1:|S|}, b_{1:|\alpha|}) \times \mathcal{N}(b_{|\alpha|+1}; f_{v_1}(a_{1:|S|}), f_{v_2}(a_{1:|S|})),$$

$$[\![(S, V, \alpha) \vdash_2 (v_0 := \texttt{if } (v_1 > v_2) \ v_3 \ \texttt{else } v_4) : (S, V \cup \{v_0\}, \alpha)]\!](p, f, l) = (p, f \oplus f'_{v_0}, l)$$
$$(\text{where } f'_{v_0}(a_{1:|S|}) = \text{if } (f_{v_1}(a_{1:|S|}) > f_{v_2}(a_{1:|S|})) \text{ then } f_{v_3}(a_{1:|S|}) \text{ else } f_{v_4}(a_{1:|S|})),$$

$$[\![(S, V, \alpha) \vdash_2 (v_0 := r) : (S, V \cup \{v_0\}, \alpha)]\!](p, f, l) = (p, f \oplus f'_{v_0}, l)$$
$$(\text{where } f'_{v_0}(a_{1:|S|}) = r),$$

$$[\![(S, V, \alpha) \vdash_2 (v_0 := v_1) : (S, V \cup \{v_0\}, \alpha)]\!](p, f, l) = (p, f \oplus f'_{v_0}, l)$$
$$(\text{where } f'_{v_0}(a_{1:|S|}) = f_{v_1}(a_{1:|S|})),$$

$$[\![(S, V, \alpha) \vdash_2 (v_0 := p'(v_0, v_1)) : (S, V \cup \{v_0\}, \alpha)]\!](p, f, l) = (p, f \oplus f'_{v_0}, l)$$
$$(\text{where } f'_{v_0}(a_{1:|S|}) = [\![p']\!](f_{v_0}(a_{1:|S|}), f_{v_1}(a_{1:|S|}))).$$

Finally, we define $p_C$ for the well-initialised well-typed programs $C$, i.e., programs $C$ for which we can derived

$$([], \emptyset, []) \vdash_1 C : (S, V, \alpha).$$

For such a $C$, the definition of $p_C$ is given below:

$$p_C(z_{1:|S|}) = p(z_{1:|S|}) \times l(z_{1:|S|}, \alpha)$$

where $(p, \_, l) = [\![([], \emptyset, []) \vdash_1 C : (S, V, \alpha)]\!](p_0, f_0, l_0)$ for the constant-1 functions $p_0$ and $l_0$ of appropriate types and the empty family $f_0$ of functions.

## D  MARGINAL LIKELIHOOD COMPUTATION: DERIVATION AND CORRECTNESS

Let $x_n$ be the random variable (RV) that is observed by the command $\texttt{obs}(\mathcal{N}(v_0, v_1), r)$ and $x_{1:(n-1)}$ be the $(n-1)$ RVs that are observed before the command. When our algorithm is about to analyse this observe command, we have (an estimate of) $p(x_{1:(n-1)})$ by induction. Then, the marginal likelihood of $x_{1:n}$ can be computed as follows:

$$p(x_{1:(n-1)}, x_n)$$
$$= \iint p(x_{1:(n-1)}, x_n, v_0, v_1) \, dv_0 \, dv_1$$
$$= \iint p(x_{1:(n-1)}) \, p(v_0, v_1 | x_{1:(n-1)}) \, p(x_n | x_{1:(n-1)}, v_0, v_1) \, dv_0 \, dv_1$$
$$\approx p(x_{1:(n-1)}) \iint p_h(v_0, v_1) \, p(x_n | x_{1:(n-1)}, v_0, v_1) \, dv_0 \, dv_1$$
$$\qquad \text{// The filtering distribution } p(v_0, v_1 | x_{1:(n-1)}) \text{ is approximated by } p_h.$$
$$= p(x_{1:(n-1)}) \iint p_h(v_0, v_1) \, p(x_n | v_0, v_1) \, dv_0 \, dv_1$$
$$\qquad \text{// The RV } x_n \text{ is conditionally independent of } x_{1:(n-1)} \text{ given } v_0, v_1.$$
$$= p(x_{1:(n-1)}) \iint p_h(v_0, v_1) \, \mathcal{N}(r; v_0, v_1) \, dv_0 \, dv_1$$
$$\qquad \text{// } p(x_{1:(n-1)}) \text{ is } Z \text{ in the description of } \text{INFER}(A_i) \text{ in Section 4, and the neural network}$$
$$\qquad \text{// } nn_{\text{intg}, \phi_7} \text{ aims at approximating the integral term accurately.}$$

This derivation leads to the equation in the main text.

In a setting of probabilistic programming where observations are allowed to be different in true and false branches, the marginal likelihood may fail to be defined, and such a setting is beyond the scope of our language. Using variables multiple times or having observe commands spread out in the program does not make differences in the derivation above.

Table 2: Full list of the model classes in the empirical evaluation.

| Section | Model class | Description | Detail |
|---------|-------------|-------------|--------|
| §5.1 | gauss | Gaussian models with a latent variable and an observation where the mean of the Gaussian likelihood is an affine transformation of the latent. | Appendix E.1.1 |
| | hierl | Hierarchical models with three hierarchically structured latent variables. | Appendix E.1.2 |
| | hierd | Hierarchical or multi-level models with both latent variables and data structured hierarchically where data are modelled as a regression of latent variables of different levels. | Appendix E.1.3 |
| | cluster | Clustering models where five observations are clustered into two groups. | Appendix E.1.4 |
| | milky and milkyo | Milky Way models, and their multiple-observations extension where five observations are made for each satellite galaxy. | Appendix E.1.5 |
| | rb | Models with the Rosenbrock function, which is expressed as an external procedure. | Appendix E.1.6 |
| §5.2 | ext1 | Models with three Gaussian variables and one deterministic variable storing the value of the function $\mathrm{nl}(x) = 50/\pi \times \arctan(x/10)$, where the models have 12 different types — four different dependency graphs of the variables, and three different positions of the deterministic $\mathrm{nl}$ variable for each of these graphs. | Appendix E.2.1 (and Fig. 10) |
| | ext2 | Models with six Gaussian variables and one $\mathrm{nl}$ variable, which are grouped into five model types based on their dependency graphs. | Appendix E.2.2 (and Fig. 11) |
| §5.3 | mulmod | Three-variable models where two latent variables follow normal distributions and the other stores the value of the function $\mathrm{mm}(x) = 100 \times x^3/(10 + x^4)$. The models in this class are grouped into three types defined by their dependency graphs and the positions of $\mathrm{mm}$ in the programs. | Appendix E.3 (and Fig. 12) |

# E  DETAILED DESCRIPTIONS FOR PROBABILISTIC MODELS USED IN THE EMPIRICAL EVALUATION

Table 2 shows the full list of the model classes that we considered in our empirical evaluation (§5). We detail the program specifications for the classes using the probabilistic programming language in §2, and then describe how our program generator generated programs from those classes randomly.

In the program specifications to follow, randomly-generated constants are written in the Greek alphabets ($\theta$), and latent and other program variables in the English alphabets. Also, we often use more intuitive variable names instead of using $z_i$ for latent variables and $v_i$ for the other program variables, to improve readability. When describing random generation of the parameter values, we let $\mathrm{U}(a, b)$ denote the uniform distribution whose domain is $(a, b) \subset \mathbb{R}$; we use this only for describing the random program generation process itself, not the generated programs (only normal distributions are used in our programs, with the notation $\mathcal{N}$).

## E.1  GENERALISATION TO NEW MODEL PARAMETERS AND OBSERVATIONS

This section details the model classes in §5.1.

### E.1.1  gauss

The model class is described as follows:

$$m_z := \theta_1; \; v_z := \theta_2'; \; c_1 := \theta_3; \; c_2 := \theta_4; \; v_x := \theta_5';$$
$$z_1 \sim \mathcal{N}(m_z, v_z); \; z_2 := z_1 \times c_1; \; z_3 := z_2 + c_2;$$
$$\mathsf{obs}(\mathcal{N}(z_3, v_x), o)$$

For each program of the class, our random program generator generated the parameter values as follows:

$$\theta_1 \sim \mathrm{U}(-5, 5), \; \theta_2 \sim \mathrm{U}(0, 20), \; \theta_2' = (\theta_2)^2, \; \theta_3 \sim \mathrm{U}(-3, 3)$$
$$\theta_4 \sim \mathrm{U}(-10, 10), \; \theta_5 \sim \mathrm{U}(0.5, 10), \; \theta_5' = (\theta_5)^2$$

and then generated the observation $o$ by running the program forward where the value for $z_1$ was sampled from $z_1 \sim \mathrm{U}(m_z - 2 \times \sqrt{v_z}, m_z + 2 \times \sqrt{v_z})$.

### E.1.2  hierl

The model class is described as follows:

$$m_g := \theta_1; \; v_g := \theta_2'; \; v_{t_1} := \theta_3'; \; v_{t_2} := \theta_4'; \; v_{x_1} := \theta_5';$$
$$v_{x_2} := \theta_6'; \; g \sim \mathcal{N}(m_g, v_g); \; t_1 \sim \mathcal{N}(g, v_{t_1}); \; t_2 \sim \mathcal{N}(g, v_{t_2});$$
$$\mathsf{obs}(\mathcal{N}(t_1, v_{x_1}), o_1); \; \mathsf{obs}(\mathcal{N}(t_2, v_{x_2}), o_2)$$

For each program of the class, our generator generated the parameter values as follows:

$$\theta_1 \sim \mathrm{U}(-5, 5), \; \theta_2 \sim \mathrm{U}(0, 50), \; \theta_2' = (\theta_2)^2, \; \theta_3 \sim \mathrm{U}(0, 10)$$
$$\theta_3' = (\theta_3)^2, \; \theta_4 \sim \mathrm{U}(0, 10), \; \theta_4' = (\theta_4)^2, \; \theta_5 \sim \mathrm{U}(0.5, 10)$$
$$\theta_5' = (\theta_5)^2, \; \theta_6 \sim \mathrm{U}(0.5, 10), \; \theta_6' = (\theta_6)^2$$

and then generated the observations $o_1$ and $o_2$ by running the program (i.e., simulating the model) forward.

### E.1.3  hierd

The model class is described as follows:

$$m_{a_0} := \theta_1; \; v_{a_0} := \theta_2'; \; v_{a_1} := \theta_3'; \; v_{a_2} := \theta_4'; \; m_b := \theta_5;$$
$$v_b := \theta_6'; \; d_1 = \theta_7; \; d_2 = \theta_8; \; v_{x_1} := \theta_9'; \; v_{x_2} := \theta_{10}';$$

$$a_0 \sim \mathcal{N}(m_{a_0}, v_{a_0}); \; a_1 \sim \mathcal{N}(a_0, v_{a_1}); \; a_2 \sim \mathcal{N}(a_0, v_{a_2});$$
$$b \sim \mathcal{N}(m_b, v_b);$$
$$t_1 := b \times d_1; \; t_2 := a_1 + t_1; \; \mathtt{obs}(\mathcal{N}(t_2, v_{x_1}), o_1);$$
$$t_3 := b \times d_2; \; t_4 := a_2 + t_3; \; \mathtt{obs}(\mathcal{N}(t_4, v_{x_2}), o_2)$$

For each program of the class, our generator generated the parameter values as follows:

$$\theta_1 \sim \mathrm{U}(-10, 10), \; \theta_2 \sim \mathrm{U}(0, 100), \; \theta_2' = (\theta_2)^2, \; \theta_3 \sim \mathrm{U}(0, 10)$$
$$\theta_3' = (\theta_3)^2, \; \theta_4 \sim \mathrm{U}(0, 10), \; \theta_4' = (\theta_4)^2, \; \theta_5 \sim \mathrm{U}(-5, 5)$$
$$\theta_6 \sim \mathrm{U}(0, 10), \; \theta_6' = (\theta_6)^2, \; \theta_7 \sim \mathrm{U}(-5, 5), \; \theta_8 \sim \mathrm{U}(-5, 5)$$
$$\theta_9 \sim \mathrm{U}(0.5, 10), \; \theta_9' = (\theta_9)^2, \; \theta_{10} \sim \mathrm{U}(0.5, 10), \; \theta_{10}' = (\theta_{10})^2$$

and then generated the observations $o_1$ and $o_2$ by running the program forward where the values for $a_0$, $a_1$, $a_2$, and $b$ in this specific simulation were sampled as follows:

$$a_0 \sim \mathrm{U}(m_{a_0} - 2 \times \sqrt{v_{a_0}}, \; m_{a_0} + 2 \times \sqrt{v_{a_0}})$$
$$a_1 \sim \mathrm{U}(a_0 - 2 \times \sqrt{v_{a_1}}, \; a_0 + 2 \times \sqrt{v_{a_1}})$$
$$a_2 \sim \mathrm{U}(a_0 - 2 \times \sqrt{v_{a_2}}, \; a_0 + 2 \times \sqrt{v_{a_2}})$$
$$b \sim \mathrm{U}(m_b - 2 \times \sqrt{v_b}, \; m_b + 2 \times \sqrt{v_b})$$

### E.1.4 cluster

The model class is described as follows:

$$m_{g_1} := \theta_1; \; v_{g_1} := \theta_2'; \; m_{g_2} := \theta_3; \; v_{g_2} := \theta_4'; \; v_x := \theta_5';$$
$$g_1 \sim \mathcal{N}(m_{g_1}, v_{g_1}); \; g_2 \sim \mathcal{N}(m_{g_2}, v_{g_2});$$
$$zero := 0; \; hund := 100;$$
$$t_1 \sim \mathcal{N}(zero, hund); \; m_1 := \mathtt{if} \; (t_1 > zero) \; g_1 \; \mathtt{else} \; g_2;$$
$$\mathtt{obs}(\mathcal{N}(m_1, v_x), o_1);$$
$$t_2 \sim \mathcal{N}(zero, hund); \; m_2 := \mathtt{if} \; (t_2 > zero) \; g_1 \; \mathtt{else} \; g_2;$$
$$\mathtt{obs}(\mathcal{N}(m_2, v_x), o_2);$$
$$t_3 \sim \mathcal{N}(zero, hund); \; m_3 := \mathtt{if} \; (t_3 > zero) \; g_1 \; \mathtt{else} \; g_2;$$
$$\mathtt{obs}(\mathcal{N}(m_3, v_x), o_3);$$
$$t_4 \sim \mathcal{N}(zero, hund); \; m_4 := \mathtt{if} \; (t_4 > zero) \; g_1 \; \mathtt{else} \; g_2;$$
$$\mathtt{obs}(\mathcal{N}(m_4, v_x), o_4);$$
$$t_5 \sim \mathcal{N}(zero, hund); \; m_5 := \mathtt{if} \; (t_5 > zero) \; g_1 \; \mathtt{else} \; g_2;$$
$$\mathtt{obs}(\mathcal{N}(m_5, v_x), o_5)$$

For each program of the class, our generator generated the parameter values as follows:

$$\theta_1 \sim \mathrm{U}(-15, 15), \; \theta_2 \sim \mathrm{U}(0.5, 50), \; \theta_2' = (\theta_2)^2$$
$$\theta_3 \sim \mathrm{U}(-15, 15), \; \theta_4 \sim \mathrm{U}(0.5, 50), \; \theta_4' = (\theta_4)^2$$
$$\theta_5 \sim \mathrm{U}(0.5, 10), \; \theta_5' = (\theta_5)^2$$

and then generated the observations $o_{1:5}$ by running the program forward.

### E.1.5 milky AND milkyo

The model class milky is described as follows:

$$m_{mass} := \theta_1; \; v_{mass} := \theta_2'; \; c_1 := \theta_3; \; v_{g_1} := \theta_4'; \; c_2 := \theta_5;$$
$$v_{g_2} := \theta_6'; \; v_{x_1} := \theta_7'; \; v_{x_2} := \theta_8';$$
$$mass \sim \mathcal{N}(m_{mass}, v_{mass});$$
$$mass_1 := mass \times c_1; \; g_1 \sim \mathcal{N}(mass_1, v_{g_1});$$
$$mass_2 := mass + c_2; \; g_2 \sim \mathcal{N}(mass_2, v_{g_2});$$

$$\texttt{obs}(\mathcal{N}(g_1, v_{x_1}), o_1); \ \texttt{obs}(\mathcal{N}(g_2, v_{x_2}), o_2)$$

For each program of milky, our generator generated the parameter values as follows:

$$\theta_1 \sim \mathrm{U}(-10, 10), \ \theta_2 \sim \mathrm{U}(0, 30), \ \theta_2' = (\theta_2)^2, \ \theta_3 \sim \mathrm{U}(-2, 2)$$
$$\theta_4 \sim \mathrm{U}(0, 10), \ \theta_4' = (\theta_4)^2, \ \theta_5 \sim \mathrm{U}(-5, 5), \ \theta_6 \sim \mathrm{U}(0, 10)$$
$$\theta_6' = (\theta_6)^2, \ \theta_7 \sim \mathrm{U}(0.5, 10), \ \theta_7' = (\theta_7)^2, \ \theta_8 \sim \mathrm{U}(0.5, 10)$$
$$\theta_8' = (\theta_8)^2$$

and then generated the observations $o_1$ and $o_2$ by running the program forward.

Everything remained the same for the milkyo class, except that the two obs commands were extended to $\texttt{obs}(\mathcal{N}(g_1, v_{x_1}), [o_1, o_2, o_3, o_4, o_5])$ and $\texttt{obs}(\mathcal{N}(g_2, v_{x_2}), [o_6, o_7, o_8, o_9, o_{10}])$, respectively, and all the observations were generated similarly by running the extended model forward.

### E.1.6 rb

The model class rb is described as follows:

$$m_{z_1} := \theta_1; \ v_{z_1} := \theta_2'; \ m_{z_2} := \theta_3; \ v_{z_2} := \theta_4'; \ v_x := \theta_5';$$
$$z_1 \sim \mathcal{N}(m_{z_1}, v_{z_1}); \ z_2 \sim \mathcal{N}(m_{z_2}, v_{z_2}); \ r := \mathrm{Rosenbrock}(z_1, z_2);$$
$$\texttt{obs}(\mathcal{N}(r, v_x), o)$$

where $\mathrm{Rosenbrock}(z_1, z_2) = 0.05 \times (z_1 - 1)^2 + 0.005 \times (z_2 - z_1{}^2)^2$. For each program of the class, our generator generated the parameter values as follows:

$$\theta_1 \sim \mathrm{U}(-8, 8), \ \theta_2 \sim \mathrm{U}(0, 5), \ \theta_2' = (\theta_2)^2, \ \theta_3 \sim \mathrm{U}(-8, 8)$$
$$\theta_4 \sim \mathrm{U}(0, 5), \ \theta_4' = (\theta_4)^2, \ \theta_5 \sim \mathrm{U}(0.5, 10), \ \theta_5' = (\theta_5)^2$$

and then generated the observation $o$ by running the program forward where the values for $z_1$ and $z_2$ in this specific simulation were sampled as follows:

$$z_1 \sim \mathrm{U}(m_{z_1} - 1.5 \times \sqrt{v_{z_1}}, \ m_{z_1} + 1.5 \times \sqrt{v_{z_1}})$$
$$z_2 \sim \mathrm{U}(m_{z_2} - 1.5 \times \sqrt{v_{z_2}}, \ m_{z_2} + 1.5 \times \sqrt{v_{z_2}})$$

### E.2 GENERALISATION TO NEW MODEL STRUCTURES

This section details the model classes, and different types in each model class in §5.2. For readability, we present canonicalised dependency graphs where variables are named in the breadth-first order. In the experiments reported in this section, we used a minor extension of our probabilistic programming language with procedures taking one parameter.

### E.2.1 ext1

Fig. 10 shows the dependency graphs for all model types in ext1. The variables $z_0, z_1, \ldots$ and $x_1, x_2, \ldots$ represent latent and observed variables, respectively, and observed variables are colored in gray. The red node in each graph represents the position of the nl variable.

Our program generator in this case generates programs from the whole model class ext1; it generates programs of all twelve different types in ext1. We explain this generation process for the model type (1,1) in Fig. 10, while pointing out that the similar process is applied to the other eleven types. To generate programs of the model type (1,1), we use the following program template:

$$m_{z_0} := \theta_1; \ v_{z_0} := \theta_2'; \ v_{z_2} := \theta_3'; \ v_{z_3} := \theta_4'; \ v_{x_1} := \theta_5';$$
$$z_0 \sim \mathcal{N}(m_{z_0}, v_{z_0}); \ z_1 := \mathrm{nl}(z_0); \ z_2 \sim \mathcal{N}(z_1, v_{z_2}); \ z_3 \sim \mathcal{N}(z_2, v_{z_3});$$
$$\texttt{obs}(\mathcal{N}(z_3, v_{x_1}), o_1)$$

where $\mathrm{nl}(z) = 50/\pi \times \arctan(z/10)$. The generation involves randomly sampling the parameters of this template, converting the template into a program in our language, and creating synthetic observations. Specifically, our generator generates the parameter values as follows:

$$\theta_1 \sim \mathrm{U}(-5, 5), \ \theta_2 \sim \mathrm{U}(0, 20), \ \theta_2' = (\theta_2)^2, \ \theta_3 \sim \mathrm{U}(0, 20), \ \theta_3' = (\theta_3)^2$$

$$\theta_4 \sim \mathrm{U}(0, 20),\ \theta_4' = (\theta_4)^2,\ \theta_5 \sim \mathrm{U}(0.5, 10),\ \theta_5' = (\theta_5)^2$$

and generates the observation $o_1$ by running the program forward where the values for $z_{0:3}$ in this specific simulation were sampled (and fixed to specific values) as follows:

$$z_0 \sim \mathrm{U}(m_{z_0} - 2 \times \sqrt{v_{z_0}},\ m_{z_0} + 2 \times \sqrt{v_{z_0}})$$
$$z_1 = \mathrm{nl}(z_0)$$
$$z_2 \sim \mathrm{U}(z_1 - 2 \times \sqrt{v_{z_2}},\ z_1 + 2 \times \sqrt{v_{z_2}})$$
$$z_3 \sim \mathrm{U}(z_2 - 2 \times \sqrt{v_{z_3}},\ z_2 + 2 \times \sqrt{v_{z_3}}).$$

The generator uses different templates for the other eleven model types in ext1, while sharing the similar process for generation of the parameters and observations.

### E.2.2 ext2

Fig. 11 shows the dependency graphs for all five model types in ext2. Programs of these five types are randomly generated by our program generator. As in the ext1 case, we explain the generator only for one model type, which corresponds to the first dependency graph in Fig. 11. To generate programs of this type, we use the following program template:

$$m_{z_0} := \theta_1;\ v_{z_0} := \theta_2';\ v_{z_1} := \theta_3';\ v_{z_3} := \theta_4';\ v_{z_4} := \theta_5';\ v_{z_5} := \theta_6';\ v_{z_6} := \theta_7';$$
$$v_{x_1} := \theta_8';\ v_{x_2} := \theta_9';\ v_{x_3} := \theta_{10}';\ v_{x_4} := \theta_{11}';$$
$$z_0 \sim \mathcal{N}(m_{z_0}, v_{z_0});\ z_1 \sim \mathcal{N}(z_0, v_{z_1});\ z_2 := \mathrm{nl}(z_0);\ z_3 \sim \mathcal{N}(z_0, v_{z_3});$$
$$z_4 \sim \mathcal{N}(z_1, v_{z_4});\ z_5 \sim \mathcal{N}(z_1, v_{z_5});\ z_6 \sim \mathcal{N}(z_2, v_{z_6});$$
$$\mathsf{obs}(\mathcal{N}(z_4, v_{x_1}), o_1);\ \mathsf{obs}(\mathcal{N}(z_5, v_{x_2}), o_2);\ \mathsf{obs}(\mathcal{N}(z_6, v_{x_3}), o_3);\ \mathsf{obs}(\mathcal{N}(z_3, v_{x_4}), o_4)$$

In order to generate a program of this model type and observations, our generator instantiates the parameters of the template as follows:

$$\theta_1 \sim \mathrm{U}(-5, 5),\ \theta_2 \sim \mathrm{U}(0, 10),\ \theta_2' = (\theta_2)^2,\ \theta_3 \sim \mathrm{U}(0, 10),\ \theta_3' = (\theta_3)^2,\ \theta_4 \sim \mathrm{U}(0, 10),\ \theta_4' = (\theta_4)^2$$
$$\theta_5 \sim \mathrm{U}(0, 10),\ \theta_5' = (\theta_5)^2,\ \theta_6 \sim \mathrm{U}(0, 10),\ \theta_6' = (\theta_6)^2,\ \theta_7 \sim \mathrm{U}(0, 10),\ \theta_7' = (\theta_7)^2$$
$$\theta_8 \sim \mathrm{U}(0, 10),\ \theta_8' = (\theta_8)^2,\ \theta_9 \sim \mathrm{U}(0, 10),\ \theta_9' = (\theta_9)^2,\ \theta_{10} \sim \mathrm{U}(0, 10),\ \theta_{10}' = (\theta_{10})^2$$
$$\theta_{11} \sim \mathrm{U}(0, 10),\ \theta_{11}' = (\theta_{11})^2.$$

Then, it generates the observations $o_{1:4}$ by running the program forward where the values for $z_{0:6}$ in this specific simulation were sampled (and fixed to specific values) as follows:

$$z_0 \sim \mathrm{U}(m_{z_0} - 2 \times \sqrt{v_{z_0}},\ m_{z_0} + 2 \times \sqrt{v_{z_0}})$$
$$z_1 \sim \mathrm{U}(z_0 - 2 \times \sqrt{v_{z_1}},\ z_0 + 2 \times \sqrt{v_{z_1}})$$
$$z_2 = \mathrm{nl}(z_0)$$
$$z_3 \sim \mathrm{U}(z_0 - 2 \times \sqrt{v_{z_3}},\ z_0 + 2 \times \sqrt{v_{z_3}})$$
$$z_4 \sim \mathrm{U}(z_1 - 2 \times \sqrt{v_{z_4}},\ z_1 + 2 \times \sqrt{v_{z_4}})$$
$$z_5 \sim \mathrm{U}(z_1 - 2 \times \sqrt{v_{z_5}},\ z_1 + 2 \times \sqrt{v_{z_5}})$$
$$z_6 \sim \mathrm{U}(z_2 - 2 \times \sqrt{v_{z_6}},\ z_2 + 2 \times \sqrt{v_{z_6}}).$$

The generator uses different templates for the other four model types in ext2, while sharing the similar process for generation of the parameters and observations.

### E.3 Test-time efficiency in comparison with alternatives

This section details the mulmod class in §5.3, which has three different model types. Fig. 12 shows the dependency graphs for all the model types. The red node in each graph represents the position of the mm variable. We used all the three types in training, applied the learnt inference algorithm to programs in the third model type, and compared the results with those returned by HMC.

We similarly explain the generator only using the model type corresponding to the first dependency graph in Fig. 12. To generate programs of this type, we use the following program template:

$$m_{z_0} := \theta_1;\ v_{z_0} := \theta_2';\ v_{z_1} := \theta_3';\ v_{x_1} := \theta_4';$$

$$z_0 \sim \mathcal{N}(m_{z_0}, v_{z_0}); \ z_1 \sim \mathcal{N}(z_0, v_{z_1}); \ z_2 := \text{mm}(z_1); \ \text{obs}(\mathcal{N}(z_2, v_{x_1}), o_1)$$

where $\text{mm}(x) = 100 \times x^3/(10 + x^4)$. For each program in this model type, our generator instantiates the parameter values as follows:

$$\theta_1 \sim \text{U}(-5, 5), \ \theta_2 \sim \text{U}(0, 20), \ \theta'_2 = (\theta_2)^2, \ \theta_3 \sim \text{U}(0, 20), \ \theta'_3 = (\theta_3)^2$$
$$\theta_4 \sim \text{U}(0.5, 10), \ \theta'_4 = (\theta_4)^2$$

and synthesises the observation $o_1$ by running the program forward where the values for $z_{0:2}$ in this specific simulation were sampled (and fixed to specific values) as follows:

$$z_0 \sim \text{U}(m_{z_0} - 2 \times \sqrt{v_{z_0}}, \ m_{z_0} + 2 \times \sqrt{v_{z_0}})$$
$$z_1 \sim \text{U}(z_0 - 2 \times \sqrt{v_{z_1}}, \ z_0 + 2 \times \sqrt{v_{z_1}})$$
$$z_2 = \text{mm}(z_1).$$

The generator uses different templates for the other two model types in mulmod, while sharing the similar process for instantiation of the parameters and observations.

## F  DETAILED EVALUATION SETUP

In our evaluation, the dimension $s$ of the internal state $h$ was 10 (i.e., $h \in \mathbb{R}^{10}$). We used the same neural network architecture for all the neural network components of our inference algorithm INFER. Each neural network had three linear layers and used the $\tanh$ activation. The hidden dimension was 10 for each layer in all the neural networks except for $nn_{\text{de}}$ where the hidden dimensions were 50. The hyper-parameter in our optimisation objective (§4) was set to $\lambda = 2$ in the evaluation. For HMC, we used the NUTS sampler (Hoffman & Gelman, 2014). We did not use GPUs.

Before running our inference algorithm, we canonicalise the names of variables in a given program based on its dependency (i.e., data-flow) graph. Although not perfect, this preprocessing removes a superficial difference between programs caused by different variable names, and enables us to avoid unnecessary complexity caused by variable-renaming symmetries at training and inference times.

## G  LOSSES FOR hierd, cluster, milkyo, AND rb

Fig. 13 shows the average training and test losses under three random seeds for hierd, cluster, milkyo, and rb. The later part of Fig. 13a, 13c and 13d shows cases where the test loss surges. This was when the loss of only a few programs in the test set (of 50 programs) became large. Even in this situation, the losses of the rest remained small. We give analyses for cluster and rb separately in §H.

## H  MULTIMODAL POSTERIORS: cluster AND rb

The cluster and rb classes in §5.1 posed another challenge: the models often had multimodal posteriors, and it was significantly harder for our meta-algorithm to learn an optimal inference algorithm. To make the evaluation partially feasible for rb, we changed two parts of our meta-algorithm slightly, as well as increasing the size of the test set from 50 to 100. First, we used importance samples instead of samples by HMC, which often failed to converge, to learn an inference algorithm. Second, our random program generator placed some restriction on the programs it generated (e.g., by using tight boundaries on some model parameters), guided by the analysis of the geometry of the Rosenbrock function (Pagani et al., 2019). Consequently, HMC (with 500K samples after 50K warmups) failed to converge for only one fifth of the test programs.

Fig. 14 shows the similar comparison plots between reference and predicted marginal posteriors for 10 test programs of the rb type, after 52.4K epochs. Our inference algorithm computed the posteriors precisely for most of the programs except two (pgm75 and pgm79) with significant multimodality. The latent variable pgm75_z0 had at least two modes at around $-10$ (visible in the figure) and around 10 (hidden in the figure)[6]. Our inference algorithm showed a mode-seeking behavior for this latent

---

[6]The blue reference plots were drawn using an HMC chain, but the HMC chain got stuck in the mode around $-10$ for this variable.

variable. Similarly, the variable pgm79_z0 had at least two modes in the similar domain region (one shown and one hidden), but this time our inference algorithm showed a mode-covering behavior.

The multimodality issue raises two questions. First, how can our meta-algorithm generate samples from the posterior more effectively so that it can optimise the inference algorithm for classes of models with multimodal posteriors? For example, our current results for cluster suffer from the fact that the samples used in the training are often biased (i.e., only from a single mode of the posterior). One possible direction would be to use multiple Markov chains simultaneously and apply ideas from the mixing-time research. Second, how can our white-box inference algorithm catch more information from the program description and find non-trivial properties that may be useful for computing the posterior distributions having multiple modes? We leave the answers for future work.

## I  TRAINING AND TEST LOSSES FOR THE OTHER CASES IN ext1

Fig. 15 shows the average training and test losses in the ext1 experiment runs (under three different random seeds) for generalisation to the last (4th) dependency graph and to all three positions of the nl variable.

## J  TRAINING AND TEST LOSSES FOR THE OTHER CASES IN ext2

Fig. 16 shows the average training and test losses in the ext2 experiment runs (under three different random seeds) for generalisation to the 4th and 5th dependency graphs.

## K  QUANTIFIED ACCURACY OF PREDICTED POSTERIORS

For accuracy, it would be ideal to report $\mathrm{KL}[p||q]$, where $p$ is the fully joint target posterior and $q$ is the predicted distribution. It is, however, hard to compute this quantity since often we cannot compute the density of $p$. One (less convincing) alternative is to compute $\mathrm{KL}[p'(z)||q(z)]$ for a latent variable $z$ where $p'(z)$ is the best Gaussian approximation (i.e., the best approximation using the mean and standard deviation) for the true marginal posterior $p(z)$, and average the results over all the latent variables of interest. We computed $\mathrm{KL}[p'(z)||q(z)]$ for the test programs from ext1 and ext2 in §5.2, and for the three from mulmod that are reported in §5.3.

For ext1 and ext2, we measured the average $\mathrm{KL}[p'(z)||q(z)]$ over all the latent variables $z$ in the test programs. For instance, if there were 90 test programs and each program had three latent variables, we averaged $90 \times 3 = 270$ KL measurements. In an experiment run for ext1 (which tested generalisation to an unseen dependency graph), the average KL was around $1.32$. In an experiment run for ext2, the estimation was around $0.95$. When we replaced $q$ with a normal distribution that is highly flat (with mean $0$ and standard deviation 10K), the estimation was $7.11$ and $7.57$, respectively. The results were similar in all the other experiment runs that were reported in §5.2.

For the three programs from mulmod in §5.3, $p'$ was the best Gaussian approximation whose mean and standard deviation were estimated by the reference importance sampler (IS-ref), and $q$ was either the predicted marginal posterior by the learnt inference algorithm or the best Gaussian approximation whose mean and standard deviation were estimated by HMC. The average KL was around $1.19$ when $q$ was the predicted posterior, while the estimation was $40.9$ when $q$ was the best Gaussian approximation by HMC. The results demonstrate that the predicted posteriors were more accurate on average than HMC at least in terms of $p'$.

## L  PROGRAM IN §5.3

Fig. 17 shows the program that is reported in §5.3, written in our probabilistic programming language.

## M  DISCUSSION OF THE COST OF IS-PRED VS. IS-PRIOR

Our approach (IS-pred) must scan the given program "twice" at test time, once for computing the proposal using the learnt neural networks and another for running the importance sampler with the

predicted proposal, while IS-prior only needs to scan the program once. Although it may seem that IS-prior has a huge advantage in terms of saving the wall-clock time, our observation is that the effect easily disappears as the sample size increases. In fact, going through the neural networks in our approach (i.e., the first scanning of the program) does not depend on the sample size, and so its time cost remains constant given the program; the time cost was 0.6ms for the reported test program (`pgm19`) in §5.3.

## N    LIMITATIONS AND FUTURE WORK

Currently, a learnt inference algorithm in our work does not generalise to programs with different sizes (Yan et al., 2020), e.g., from clustering models with two clusters to those with ten clusters. Each model class assumes a fixed number of variables, and the neural networks crucially exploit the assumption. Also, our meta-algorithm does not scale in practice. When applied to large programs, e.g., state-space models with a few hundred time steps, it cannot learn an optimal inference algorithm within a reasonable amount of time. Overcoming these limitations is a future work. Another direction that we are considering is to remove the strong independence assumption (via mean field Gaussian) on the approximating distribution in our inference algorithm, and to equip the algorithm with the capability of generating an appropriate form of the approximation distribution with rich dependency structure, by, e.g., incorporating the ideas from Ambrogioni et al. (2021). This direction is closely related to automatic guide generation in Pyro (Bingham et al., 2018).

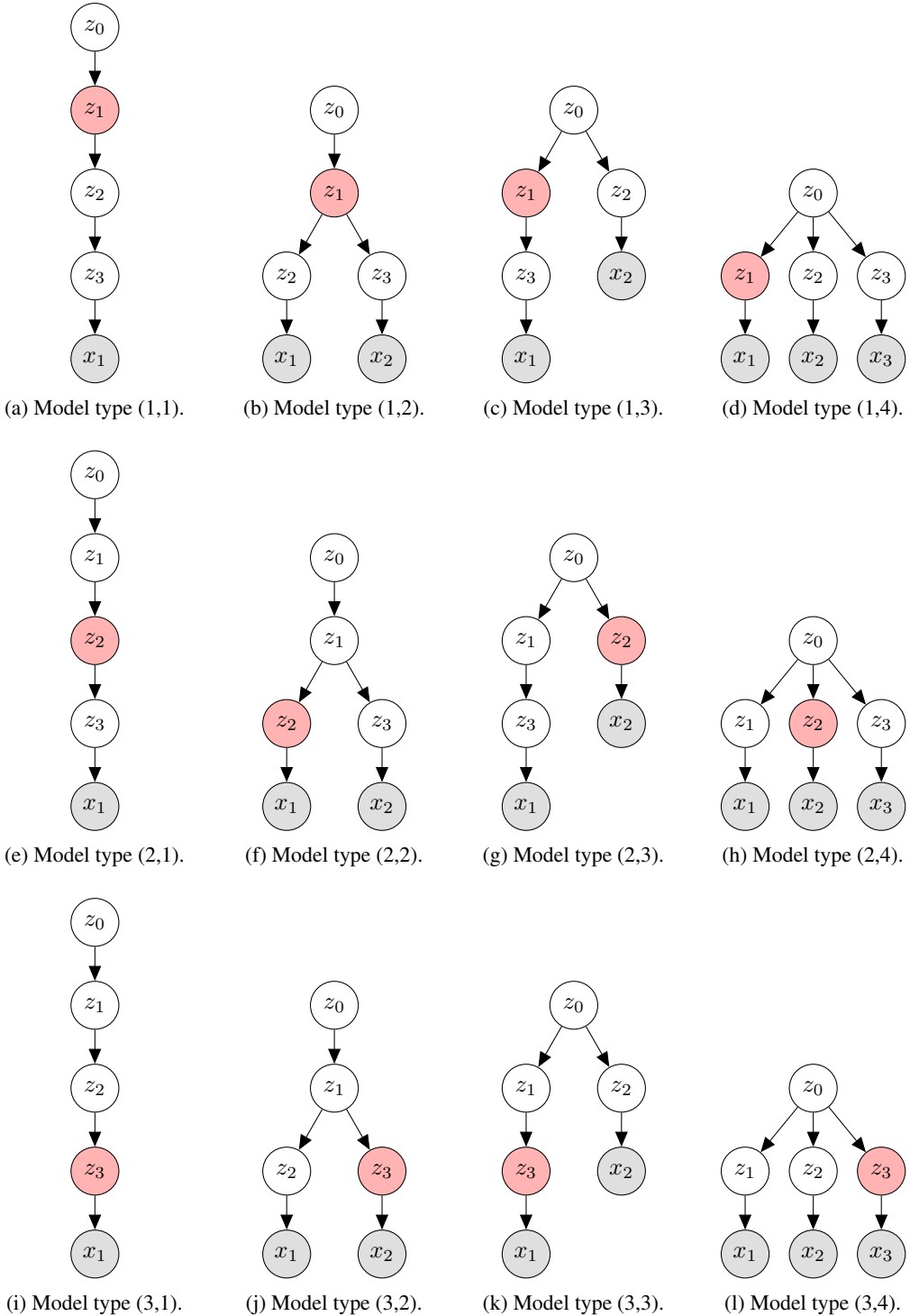

Figure 10: Canonicalised dependency graphs for all 12 model types in ext1. The rows are for different positions of the nl variable, and the columns are for different dependency graphs: the model type $(i,j)$ means one of the 12 model types in ext1 that corresponds to the $i$-th position of the nl variable and $j$-th dependency graph.

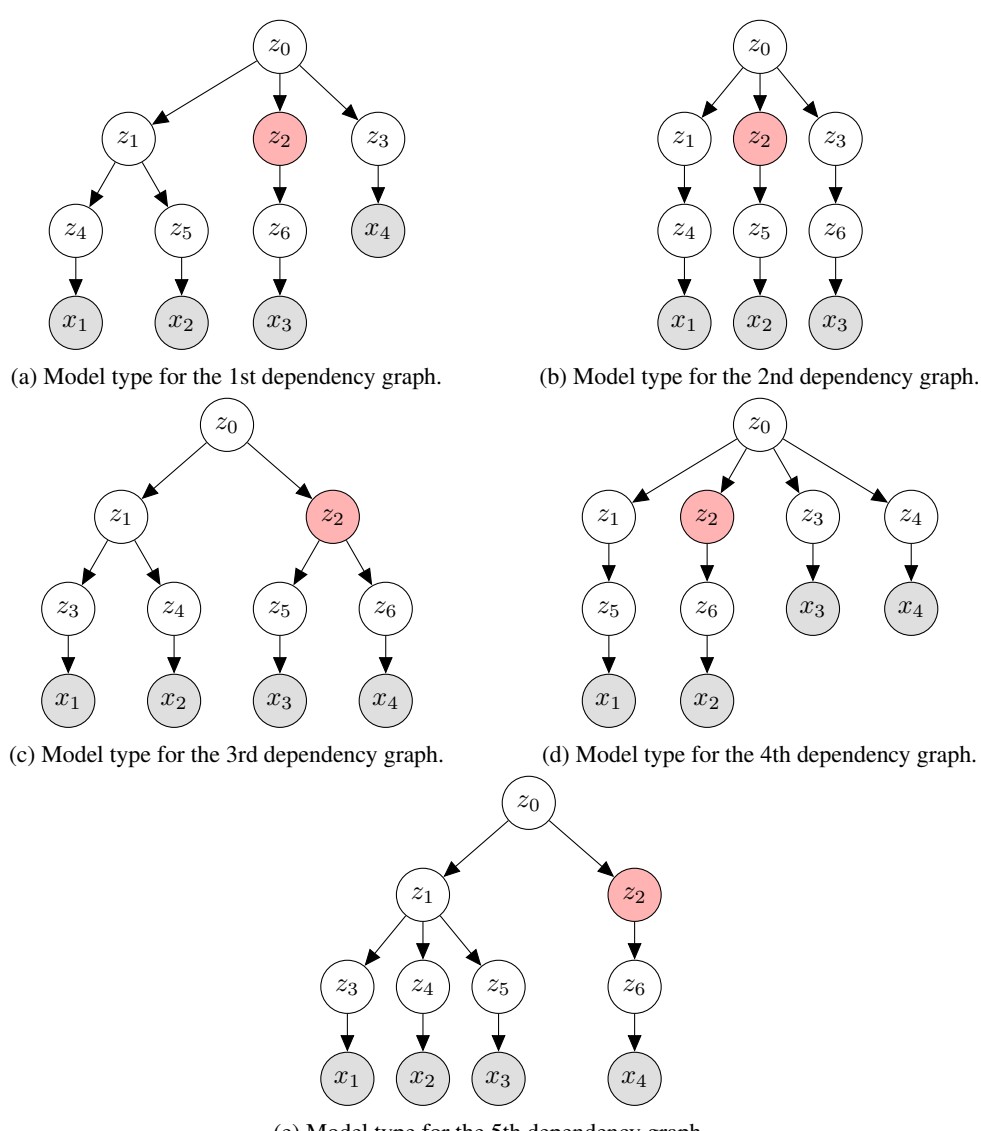

(a) Model type for the 1st dependency graph.

(b) Model type for the 2nd dependency graph.

(c) Model type for the 3rd dependency graph.

(d) Model type for the 4th dependency graph.

(e) Model type for the 5th dependency graph.

Figure 11: Canonicalised dependency graphs for all five model types in ext2.

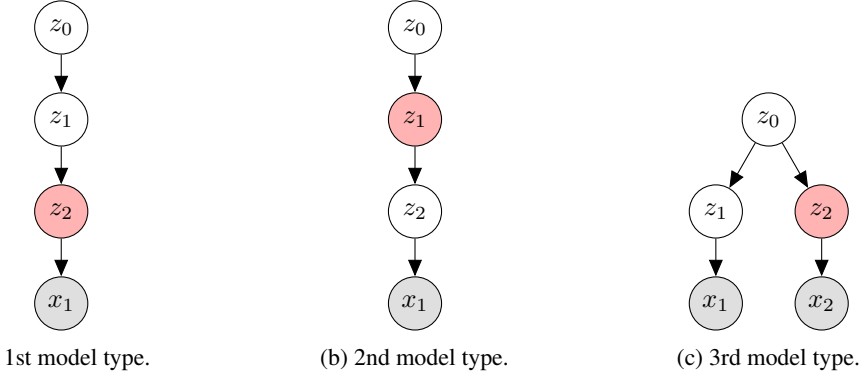

(a) 1st model type.

(b) 2nd model type.

(c) 3rd model type.

Figure 12: Canonicalised dependency graphs for all three model types in the mulmod class.

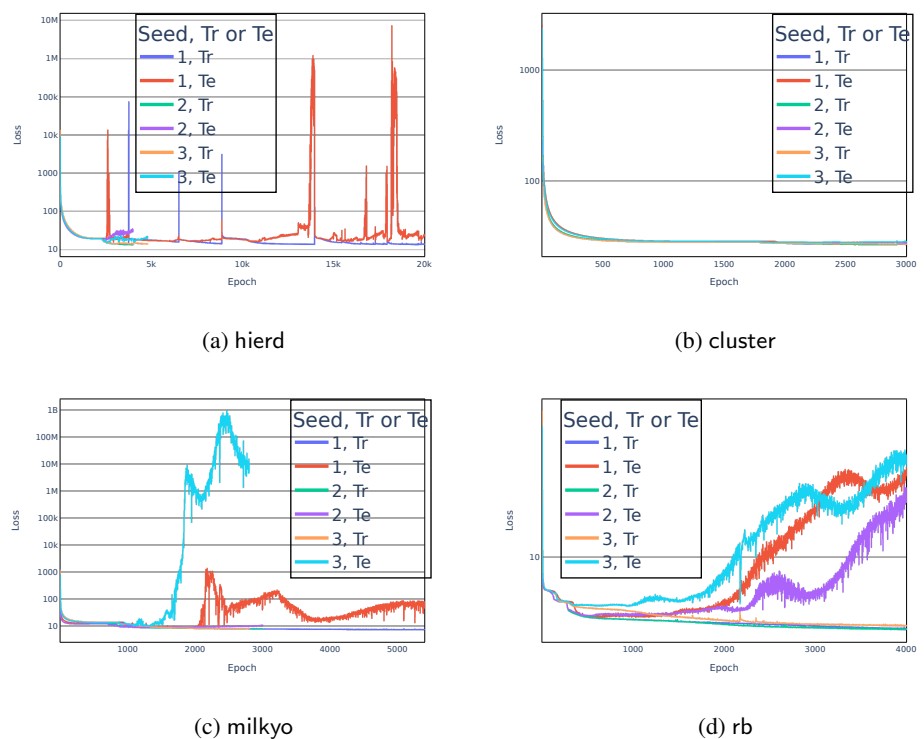

Figure 13: Losses for hierd, cluster, milkyo, and rb. The $y$-axes are log-scaled. The surges in later epochs of Fig. 13a, 13c and 13d were due to only a single or a few test programs out of 50.

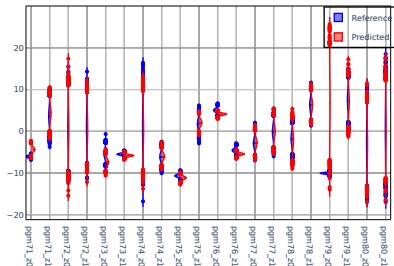

Figure 14: Comparisons of reference and predicted marginal posteriors for 10 programs in the rb test set.

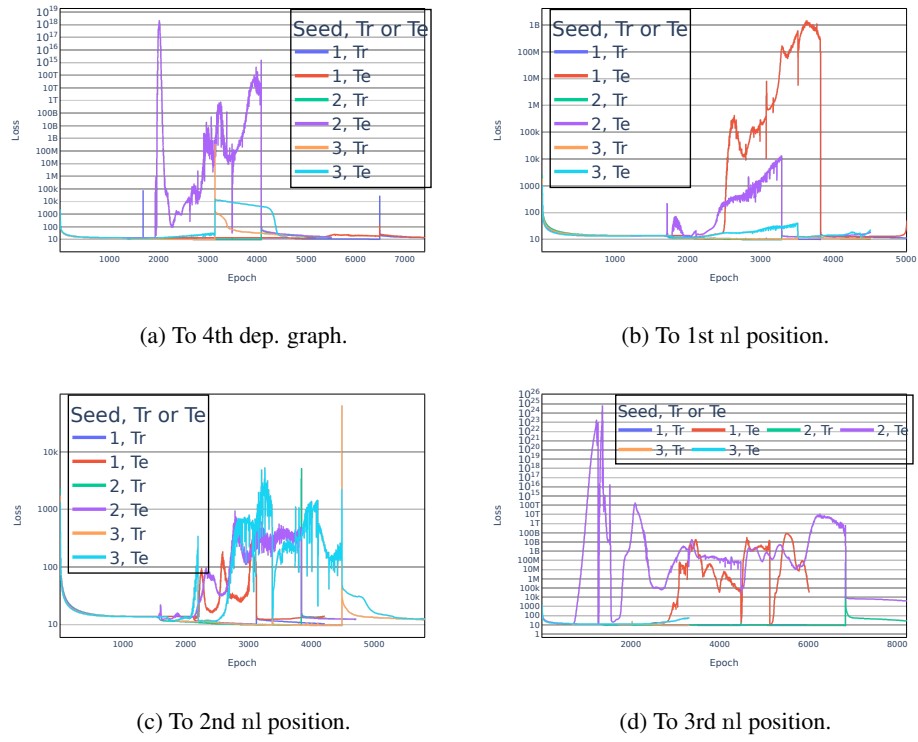

(a) To 4th dep. graph.

(b) To 1st nl position.

(c) To 2nd nl position.

(d) To 3rd nl position.

Figure 15: Average training and test losses for generalisation to the last (4th) dependency graph and to all three positions of the nl variable in ext1. The y-axes are log-scaled.

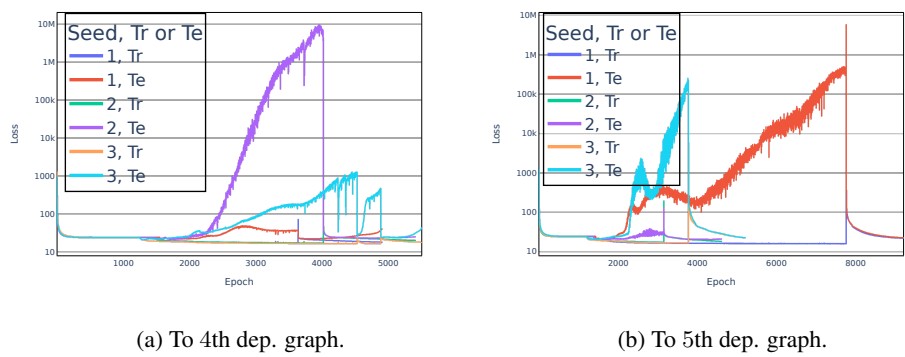

(a) To 4th dep. graph.

(b) To 5th dep. graph.

Figure 16: Average training and test losses for generalisation to the 4th and 5th dependency graphs in ext2. The y-axes are log-scaled.

$$a := 3.93;\ b := 348.16;\ c := 57.5;\ d := 14.04;\ e := 40.34;$$
$$z_1 \sim \mathcal{N}(a, b);\ z_2 \sim \mathcal{N}(z_1, c);\ z_3 := \mathrm{mm}(z_1);$$
$$\mathtt{obs}(\mathcal{N}(z_2, d), 53.97);\ \mathtt{obs}(\mathcal{N}(z_3, e), 0.12)$$

Figure 17: The program that is reported in §5.3, written in our probabilistic programming language.

