# OpenReview forum: "Meta-Learning an Inference Algorithm for Probabilistic Programs"
_ICLR.cc/2022/Conference — ICLR 2022 Submitted_

### Official Review · Reviewer_BQrG · 2021-11-02

**Correctness:** 2
**Technical Novelty And Significance:** 1
**Empirical Novelty And Significance:** 1
**Recommendation:** 3
**Confidence:** 4

**Main Review:**

The main drawback in this work is a lack of novelty. The use of neural networks to train a proposer for a model is not new. While the authors attempt to cast their work as something different than IC this doesn’t quite come out. The claim that a neural network for one program can be used for a different program even though the neural network takes a one hot encoded representation of the variables in the model is hard to see. A clear technical statement of what kind of cross model generalization is possible is needed.

The paper shows results across model structures where the dependency graph and the position of a function changes but the number of variables is the same. It is not clear why IC can’t deal with this minor variation in the same probabilistic program.

These models are so simple, how hard would it be to train ic on these models and then do inference? I would like to have seen ic results in this paper to believe that this work is different.

The language chosen is not a universal ppl. I can’t follow how this can be used as an intermediate language by a compiler for a universal ppl as claimed on page 3. Please show an example of how a program with unbounded random variables can be compiled into this language.

The inference in this paper looks like mean field variational inference. Which makes me wonder whether hmc is really such a good comparison. Please show some comparisons to vi. In Stan this would be trivial to run since you are already running hmc.

The models are very simplistic with no discrete variables and no multimodal posteriors. It is not a meaningful claim to make (footnote on page 9) that the inference algorithm provides a good coverage of the posterior by covering all the modes. The posteriors shown for multimodal models should at least look multimodal.  Variational distance or symmetric kL divergence results would be needed to make claims about correctness of the posterior.

Regarding ess per second results these can be misleading. The algorithm might have a cap of ess for example. It would be better to run your algorithm for the same duration as hmc and show higher ess numbers.


**Summary Of The Paper:**

The paper proposes a new restrictive class of probabilistic programs with fixed number of random variables and without loops. The authors then propose an inference technique that learns the parameters of a neural network for sampling from the programs posterior distribution by composing it from individual neural networks for each atomic command in the language. This technique is shown to perform well during inference once the neural network has been trained on training programs. At the very least the inference speed is shown to be very high.



**Summary Of The Review:**

The claim that the paper provides generalization of compiled inference across models is not supported by the description or the simple examples. These appear to be covered by existing work on inference compilation.

The focus on a very restricted class of ppls makes this work very limited. I don’t believe I learned anything from this paper.

---

> ### Author Response · Authors · 2021-11-22
> **Response to Reviewer BQrG**
>
> [Q1] The proposed approach seems to be just (a minor variation of) IC.
>
> As we discussed in the related work, both IC and ours (INFER) amortise the cost of repeated inference tasks. The main difference is that IC typically assumes a single model, while INFER assumes a class of models, not a single one, where models in the class may have different structures. A good analogy is the difference between a single compiled program and a compiler. The objective of IC corresponds to compiling a single program to efficient low-level code, so that the repeated applications of the code to different inputs run fast. The objective of INFER, on the other hand, corresponds to building a good compiler for a class of programs; when the compiler is given a program in the class, it should compile the program to efficient low-level code that runs fast for future inputs. Of course, we admit that this objective or ideal is not fully realised in our submission, as the reviewers pointed out. But we think that important initial ideas for this realisation have been proposed and tested in our submission.
>
> [Q2] The authors did not provide a clear technical statement about the kinds of possible cross-model generalisation (esp. under the one-hot encoding).
>
> Good point. The one-hot representation of variables is one of the major drawbacks of the proposed approach. It is not clear how much it affects training and prediction, and we believe that it limits the degree of generalisation substantially. An important consequence of the encoding is that the inference algorithm is inapplicable to a program with an arbitrary number of variables. In our ongoing work, we are trying to develop an inference algorithm that is free from the one-hot encoding and is applicable to programs with an arbitrary size.
>
> [Q3] How can a universal PPL be translated into the proposed language?
>
> We did not claim that a universal PPL can be translated into our language; we claimed the translatability from a higher-level PPL with general conditional statements and for loops into our language. As mentioned in Appendix A, programs with recursions or while loops cannot generally be translated into our language, since such programs may not terminate.

---

> > ### Comment · Reviewer_BQrG · 2021-11-29
> > **Still not clear how this differs from IC**
> >
> > The author response, "The main difference is that IC typically assumes a single model, while INFER assumes a class of models, not a single one, where models in the class may have different structures." is not convincing and doesn't address my concern. IC applies to general PPLs in which a single model can handle a very broad class of structures. As I mentioned in my review, a single probabilistic program can trivially handle the variation in structure that is presented in this paper. Hence this work is already covered under IC.
> >
> > The authors didn't respond to other technical comments pertaining to showing VI numbers or running for longer and demonstrating higher ESS. This makes me suspect that those results are not good which brings into question any minimal advancement that this paper represents.

---

### Official Review · Reviewer_mFVo · 2021-11-03

**Correctness:** 2
**Technical Novelty And Significance:** 3
**Empirical Novelty And Significance:** 1
**Recommendation:** 3
**Confidence:** 5

**Main Review:**

This work is very interesting and novel. It's a unique attempt
to learn a general inference algorithm. I think meta learning
is something that is a good fit for many Bayesian approaches
and I want to see more work like this.

I'm curious about the expressivity of the language. The
grammar suggestions a modelling language that consists of
sequence of commands. I don't see how this language would be
able to express recursive programs. It seems one would need
something like a label and jump commands to accomplish
that. This is admitted in the appendix but not really
acknowledged in the main text. I think the main paper should
reflect the present limitations and not over promise the
existing contribution.

I have some concerns about the experiments. Some examples are
fairly simple and the results for the more substantial ones
are not shown (like hierd and rb). The test losses look fairly
bad for the experiments that are shown, so I'm not fully sure
generalisation has been demonstrated. If the issue is a few bad
 generated programs maybe show median loss? The results in Figure 4
seem to be for during training, but feasibility presumably
requires similar results on unseen programs at test time?

 I also worry about correctness. As the neural networks are
used as is and not as a proposal, how much can we trust
the posteriors that come out of this method? Is there anything
that can be said about the learnt posterior? It seems right now
that the learned distribution can recover the mean of the true
parameters and maybe the variance?

Some of the writing is slightly sloppy. For example the phrase
"so-called static single assignment assumption" is used. I don't
know what that means, but I do know there is a static single
assignment intermediate representation that exists within many
compilers. I think that's what the paper meant to refer to.

Related work should cover how this approach differs from Stites et al.
(https://arxiv.org/abs/2103.00668).

**Summary Of The Paper:**

This introduces a meta-learning algorithm for learning
inference algorithms applicable to any probabilistic
program. This is accomplished by associating a neural network
with every grammar rule of a probabilistic programming
language and outputting posterior draws. This generalisation is
possible because each neural network component is feed marginal
likelihood information for each PPL instruction.


**Summary Of The Review:**

I recommend this paper for rejection. While I think the approach has the potential to work, right now it's very hard to get a sense of what was learned by the meta learning algorithm, how well any of this generalises when model structure or even observed data changes significantly, or even if the language is too restricted to make this is a significant enough contribution.

---

> ### Author Response · Authors · 2021-11-22
> **Response to Reviewer mFVo**
>
> [Q1] The main text should acknowledge explicitly the limited expressiveness of the language.
>
> We will explicitly mention in the main text that a PPL with recursion or while loops cannot be translated into our language in general, as mentioned in the appendix of the submission.
>
> [Q2] The results for more complex classes are not shown, and Figure 4 should be for “test” programs.
>
> Losses for the other program classes are shown in the Appendices G, I and J. The predicted posteriors in Figure 4 are actually for unseen test programs, not for the training programs. It is just that the inference algorithm is instantiated (using the neural network parameters) at three different training steps.
>
> [Q3] How much can we trust the predicted posterior?
>
> Please see our answer for [Q1] by Reviewer mmqX.
>
> [Q4] The authors discuss the work by Stites et al.
>
> We were unaware of the work. Thank you for the reference. We will discuss the paper in the related work section.

---

### Official Review · Reviewer_mmqX · 2021-11-03

**Correctness:** 3
**Technical Novelty And Significance:** 3
**Empirical Novelty And Significance:** 2
**Recommendation:** 5
**Confidence:** 3

**Main Review:**

- The paper claims that the learned inference algorithm works well for tasks which are "similar" to the training problems, but the notion of similarity is not fully defined, nor is there an example of how the system fails when applied to a dissimilar program. Are there diagnostics for checking the output when the model is applied to a program which is too dissimilar?
- The experimental study tests performance within a family of programs, each using small neural networks to infer each program statement. Does training on all families of programs allow the system to make accurate inferences on any of those families?  Does it allow it to generalise across families? Without some notion of how the system generalises I'm not sure when I would choose to use this rather than running HMC on my program, given a single HMC run will be faster than training the neural networks on multiple HMC runs for different programs "close" to the program of interest.
- The experiment in section 5.3 shows that the system is around 2x faster than importance sampling from the prior, but this doesn't take into account the time necessary to train the neural nets nor the time taken for the importance sampling runs used to generate the training data.
- How are losses propagated through the program? If each neural net is 3 layers, then programs with 10 statements have at most 30 layers, which is usually past the point where some amount of regularization or normalization is necessary to stabilize training or prevent vanishing gradients. Could the authors comment on the stability of training?
- What's the failure mode when the test loss diverges? Is it detectable without having HMC or other high quality samples?
- How robust is this approach to differing choices of neural net architecture?  The paper uses a 10 dimensional state space when parsing the program, but it's not clear how this value should be modified as the number of latent variables or the program complexity changes.
- Overall the paper is well written and explained, and the experimental study is detailed for the areas it covers.

**Summary Of The Paper:**

The paper presents an algorithm for composing inference algorithms out of simpler neural net building blocks, one per unique statement type in the probabilistic programming language presented. The language is simple without recursion or loops, reducing issues due to feedback problems from the approximation. The networks are trained using HMC or importance sampling samples from many programs in a similar space. There is an empirical study of several classes of small Gaussian probabilistic programs.

**Summary Of The Review:**

The paper presents a learned inference algorithm, but it's not clear how it generalises either across program types (which is necessary to amortize the training cost wrt HMC), or across neural network architectures (e.g., changing the internal state space in response to increased program complexity).  Additionally it's noted that occasionally the test error diverges, but there's no discussion of how to detect this in practice if the system was used for inference.

---

> ### Author Response · Authors · 2021-11-22
> **Response to Reviewer mmqX**
>
> [Q1] The paper should provide a clear statement about how much we can trust the predicted posterior, and about the notion of similarity among programs.
>
> In our work, two programs are similar, if they are from the same program class, but the degree of similarity varies depending on their model dynamics and observations. As you mentioned, it will be very useful to define the notions of the degree of similarity (of programs) and trustworthiness (of predicted posteriors), and extend our inference algorithm so that it returns both predictions and uncertainty of the predictions at the same time. Depending on the degree of uncertainty, it may raise an alarm. Then, the extended inference algorithm should be able to reason about not only the given program, but also the parameters of the neural networks that work on the program. Measuring the prediction uncertainty is in fact one of the goals in Bayesian deep learning, and we believe that it is a good future work to bring the ideas to our setup and make our inference algorithm more reliable.
>
> [Q2] The paper should evaluate the generalisation across program classes and across neural network architectures.
>
> Good suggestion. Though we demonstrated some cases of cross-model generalisation, the degree of generalisation that was demonstrated is quite limited. Ideally, an inference algorithm should only assume the given PPL without making additional “class” assumptions. One hurdle that we should overcome along the road to the vision is that the inference algorithm in our current setup is inapplicable to programs of an arbitrary size (or programs with an arbitrary number of variables). We are currently studying this problem in our ongoing work.
>
> In terms of generalisation to different neural network architectures (e.g., a bigger dimension of the internal states), we agree that it is useful to see how resilient our approach is as the complexity of a program and accordingly the dimension of the internal states increases. While such an empirical study may help the reader see (the limitation of) our approach better, a better scientific question might be: can we come up with an inference algorithm where neural networks only have to encode computations with a fixed complexity, in a sense that the complexity does not grow as the number of variables in a program increases? In this new setup, the other part of the inference algorithm should be able to deal with the growing complexity, and to assign simpler tasks (that are independent of the program size) to neural networks properly. This is also an important part of our ongoing work.
>
> [Q3] In Section 5.3, the “training time” for IS-pred should be considered when comparing with IS-prior.
>
> Our claim in Section 5.3 and Table 1 is the “test-time” efficiency over the alternatives, after the inference is amortised (via training) over the class of programs. When using the same number of samples at test time, it is actually the opposite; IS-pred takes more time than IS-prior because IS-pred must scan the program twice (once for computing the predicted posterior and another for running the IS with the predicted posterior as proposal) while IS-prior scans the program only once. Appendix M discusses this difference, and the additional computational cost (in IS-pred) becomes negligible as the number of IS samples increases.
>
> [Q4] How stable is the training? Is there an issue of vanishing or exploding gradients during training?
>
> We did not check this, but we agree that it is an important point to check. Even though the number of neural networks and their sizes are fixed, the size of the computation graph (dynamically built while the inference algorithm proceeds) gets bigger as the size of the program increases.

---

### Official Review · Reviewer_VGsf · 2021-11-09

**Correctness:** 3
**Technical Novelty And Significance:** 3
**Empirical Novelty And Significance:** 3
**Recommendation:** 6
**Confidence:** 2

**Main Review:**

There were quite a few terms that I was not familiar with - for example, what is the "state" of an algorithm?  I did not find a formal definition of this term in the paper.  In the definition of the INFER function, what does the keyword "in" mean?  It did not seem clear how well the white-box inference algorithm performs when the number of commands in the program is very large.

**Summary Of The Paper:**

This paper proposes an algorithm for computing an approximation of the posterior and marginal likelihood by analysing the sequence of programs using neural networks, as well as a meta-algorithm for learning the network parameters over a training set of probabilistic programs.  Experiments demonstrate the feasibility of the meta-algorithm for learning inference algorithms that generalise well to new but similar programs; these learnt algorithms were sometimes found to outperform alternatives in terms of time-efficiency.

**Summary Of The Review:**

I am not at all familiar with probabilistic programming; this does look like a serious piece of work, though I do not know how novel it is or whether the claims in the paper are correct.

---

### Author Response · Authors · 2021-11-22
**Response**

We appreciate the reviewers for their helpful comments. In this response, we focus on answering major concerns in the reviews.

---

### Decision · Program_Chairs · 2022-01-20

**Decision:**

Reject

**Comment:**

The paper presents a meta-algorithm for learning a posterior-inference algorithm for restricted probabilistic programs. While the reviews agree that this is a very interesting research direction, they also reveal that there are several questions still open. One reviewer points out that there learning to infer should take both the time for learning+inference and the generalization to other programs into account, i.e., what happens if the program is too different from the training set? Is benefit than vanishing? Moreover, as pointed out by another review, recursion as well as while loops are not yet supported. Also, the relation to IC needs some further clarification. These issues show that the paper is not yet ready for publication at ICLR. However we would like to encourage the authors to improve the work and submit it to one of the next AI venues.